# Comparison between Siliceous Concretions from the Ionian Basin and the Apulian Platform Margins (Pre-Apulian Zone), Western Greece: Implication of Differential Diagenesis on Nodules Evolution

Nicolina Bourli [1], Maria Kokkaliari [2], Nikolaos Dimopoulos [1], Ioannis Iliopoulos [2], Elena Zoumpouli [1], George Iliopoulos [3] and Avraam Zelilidis [1,*]

1   Laboratory of Sedimentology, Department of Geology, University of Patras, 26504 Patras, Greece; n_bourli@upnet.gr (N.B.); nikos.dhmopoulos@gmail.com (N.D.); zoumpouel@gmail.com (E.Z.)
2   Minerals and Rocks Research Laboratory, Department of Geology, University of Patras, 26504 Patras, Greece; kokkaliarimaria@gmail.com (M.K.); morel@upatras.gr (I.I.)
3   Laboratory of Paleontology and Stratigraphy, Department of Geology, University of Patras, 26504 Patras, Greece; iliopoulosg@upatras.gr
*   Correspondence: a.zelilidis@upatras.gr

**Abstract:** Siliceous concretions (nodules), from two different geological settings—the Apulian platform margins in Kefalonia island, and the Ionian Basin in Ithaca, Atokos, and Kastos islands—have been studied both in the field and in the laboratory. Nodule cuttings are mainly characterized by the development of a core, around which a ring (rim) has been formed. Mineralogical study, using X-ray powder diffraction (XRPD) analysis, showed that the rim is usually richer in moganite than the core. Homogeneous concretions, without discernible inner core and outer ring, were observed generally in both settings. Mineralogical analysis of the selected siliceous concretions from Kefalonia island showed the presence mostly of quartz and moganite, while calcite either was absent or participated in a few samples in minor/trace abundances. Moganite was generally abundant in all the samples from Kefalonia island. Concretions from the Ionian Basin showed a variation in the quartz, moganite, and calcite contents. Mineralogical differences were recognized both between the different studied geodynamic settings and internally in the same setting, but with different stages of development. The above-mentioned differential diagenesis on nodules evolution could be related to the presence and/or abundance of stylolites, later fluid flows, restrictions from one area to another due to synchronous fault activity, and the composition of substances dissolved in fluids. Moreover, the development of concretions produced secondary fractures in the surrounding area of the nodule-bearing rocks.

**Keywords:** Apulian platform margins; Ionian Basin; Kefalonia island; Ithaca island; Kastos island; siliceous concretions; X-ray diffraction

## 1. Introduction

Both bedded and nodular siliceous concretions are common in both deep-water and basin margins of Cretaceous limestones of the Ionian Basin (IB) and the Apulian platform margins (APM) (Pre-Apulian zone) of western Greece [1,2].

Siliceous concretions were initiated due to precursors in limestone during very early burial diagenesis and are related to redox-controlled boundaries [1,3]. As submarine landslide deposits and turbidites are common on the basin margins, there is an opportunity to examine the evolution of lithified chert in relation to synsedimentary deformation as turbidite deposition produces stable redox boundaries [4], and thus thick calciturbidites may be particularly susceptible to chert formation [1]. Younger faults that probably developed during post-Eocene compression may have provided pathways for migration of basinal fluids as salt diapirs were mobilized by overlying tectonic thickening [1].

According to Bourli et al., 2019 [1], diagenetic siliceous concretions could be formed by the replacement of other materials, such as calcium carbonate, due to the presence of high silica content in the waters that were flowing through the pre-existing stylolites within the rocks. The source of the silica is mainly biogenic, with the opaline silica of diatoms, radiolaria, and siliceous sponges being redistributed. Therefore, cherts occur as nodules; within a rock, they have different sizes, colors, textures, and structures. Nodules can progressively form nearly concentric parts, i.e., a rim, a main body, and a core. Their size ranges from a few centimeters to several tens of centimeters, and their morphology can include either subspheroidal masses or lenses. These cherts are green, grey, and black. When primary deposits exist (as many jasper and radiolarites reveal), they occur as thin beds.

According to Gül et al., 2015 [5], these siliceous concretions can be developed in limestones of different ages that can be deposited under different conditions.

The presence of stylolites—irregular dissolution surfaces, formed under pressure solutions, with variable shape and spatial distributions—characterizes most carbonate rocks, influencing potentially fluid flows. These diagenetic fluid flows have been related to stylolite networks, and furthermore it has been suggested that they were either pathways that improved permeability along their orientation (e.g., [6–8]) or obstacles to fluid flows (e.g., [7,9,10]). Stylolites in carbonates have been related to intergranular pressure solution [11,12]. Additionally, high solubility and fast reaction kinetics could be the reason they are predominantly represented in this sediment, which is more susceptible to pressure solution than other rock types. Despite their abundance in carbonate rocks, according to [13], their occurrence could be related to the fracture network. Stylolites have also been used for different approaches, including the estimation of principal compressive stress by studying their shape morphology [14] or burial compaction with vertical compression by studying their parallel bedding [15].

The aim of this work is to establish the diagenetic processes of siliceous concretions, during the Cretaceous to Palaeocene, of the Ionian Basin (IB) and the Apulian platform margins (APM) (Pre-Apulian zone). The following methodology is supported by field observations, X-ray diffraction analysis of zoned siliceous concretions, and a comparison between rocks of different ages and palaeogeographic settings, with pre-existing results from other areas of the Ionian Basin. By studying sedimentary facies and carbonate rock components and taking into account the shape of stylolites, an attempt was made to clarify the role of stylolite on differential diagenesis of siliceous concretion development. Finally, the role of siliceous concretion was studied in relation to the development of secondary fractions network.

## 2. Geological Setting of the Studied Sections

From the Triassic to the Late Cretaceous, Western Greece was part of the Apulian continental block on the southern passive margin of Tethys [16]. Widely exposed Cretaceous limestones of the (APM) (Pre-Apulian zone) characterize the island of Kefalonia and the external Ionian sub-basin characterizes Ithaca, Atokos, and Kastos islands (Figure 1).

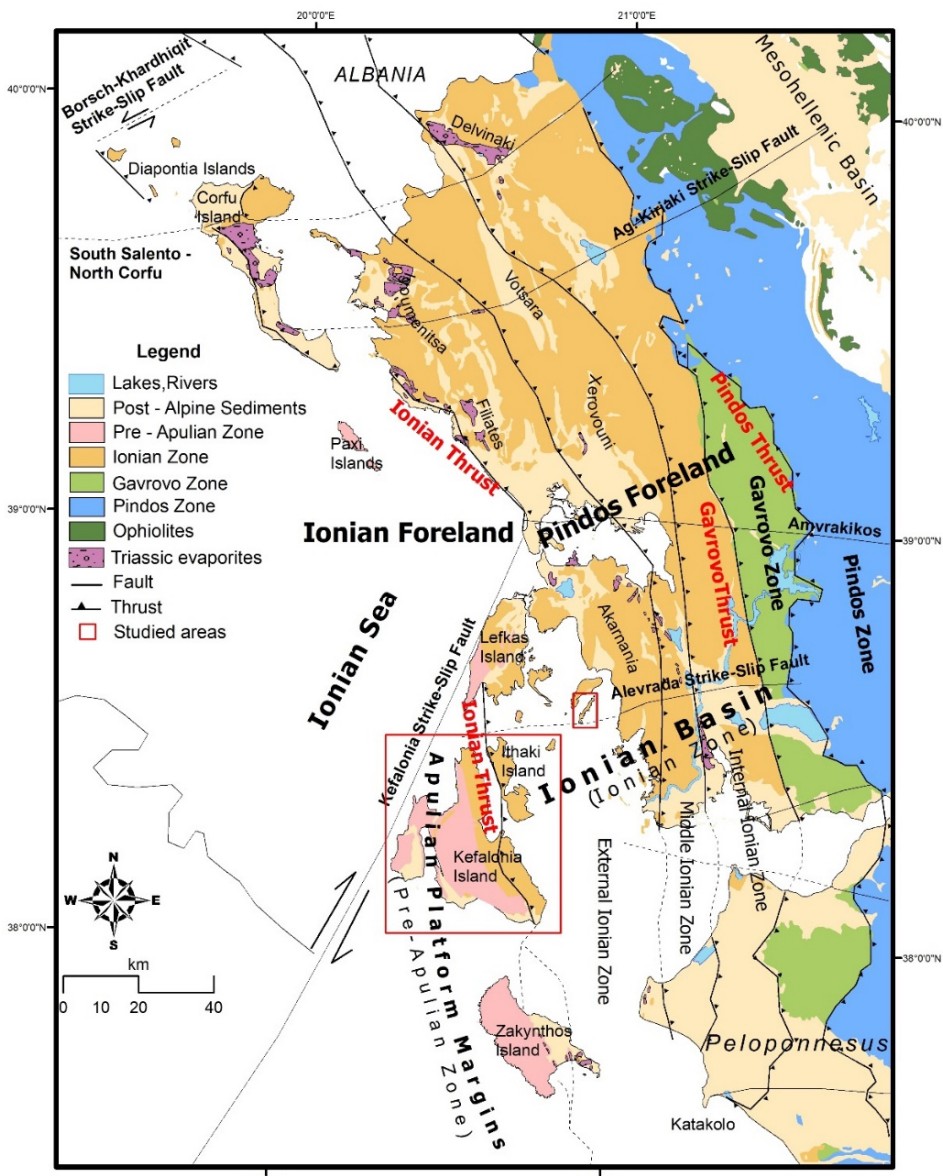

**Figure 1.** Geological map of the external Hellenides in NW Greece illustrating the principal tectonostratigraphic zones: Pindos, Gavrovo, Ionian, and Pre-Apulian Zones (modified from [17]). Red boxes show the studied areas of Kefalonia, Ithaca, Atokos, and Kastos islands (Figure 3).

Kefalonia island mainly belongs to the APM. It is considered as the autochthonous foreland of the Hellenic fold-thrust belt, generally believed to have been unaffected by major shortening [18–24]. The extensional normal faults were reactivated during the compressional regime as thrust faults showing inverted faults [25].

The IB is bounded westwards by the Ionian Thrust and eastwards by the Gavrovo Thrust (Figure 1). The Pre-Apulian or Paxoi zone to the west of the Ionian basin is regarded as the eastern margin of the Apulian platform, in Albania, Croatia, and Italy, where similar rocks occur [17,26–28].

The APM consists of Triassic to Miocene deposits, mainly neritic carbonate rocks (Figure 2a). The IB comprises sedimentary formations ranging from Triassic evaporites to Jurassic-Late Eocene carbonate rocks, including minor chert and shale horizons (Figure 2b).

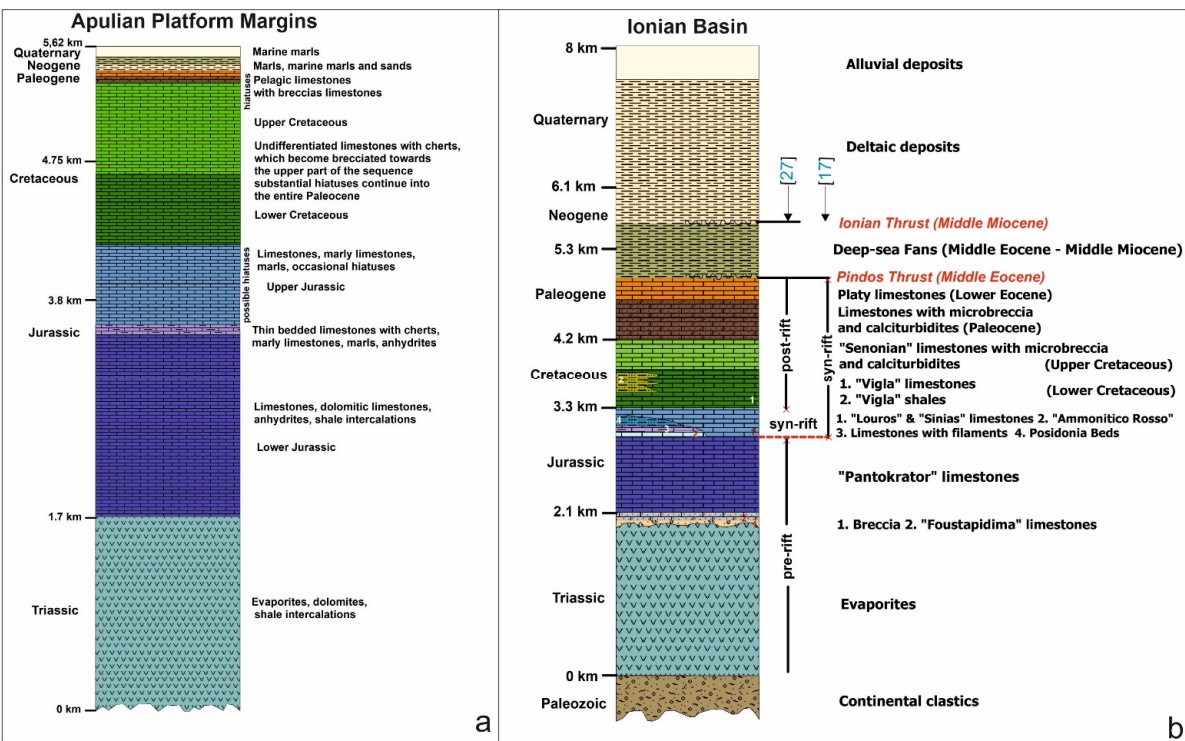

**Figure 2.** Detailed lithostratigraphic columns of (**a**) the APM (Pre-Apulian Zone) and (**b**) the IB (Ionian Zone), NW Greece (modified from [27,28]).

The accumulation of sediments in the IB took place under three different tectonic conditions [17,28]: a pre-rift stage during the Triassic to Early Jurassic, a syn-rift stage during the Middle Jurassic to Early Eocene, and a compressional stage from the Early Eocene to the Middle Miocene. During the syn-rift stage, the Ionian basin was sub-divided into sub-basins with an asymmetric geometry and different sediment thickness accumulation. Many of the normal faults during the syn-rift stage were reactivated during the compressional stage as thrusts or back-thrusts [28].

The sedimentary succession (Figure 2b) from base to top consists of the following formations: the Lower to Middle Triassic evaporates; the Upper Triassic "Foustapidima" limestones; the Lower Jurassic "Pantokrator" limestones; the Lower Jurassic pelagic "Sini-ais" limestones with their lateral equivalent; the "Louros" limestones, which pass upwards to the Lower to Upper Jurassic "Ammonitico Rosso"; the "Limestones with filaments" and the "Posidonia beds", overlying Cretaceous sedimentary rocks that comprise the Lower Cretaceous pelagic "Vigla" limestones, with their laterally equivalent "Vigla shales"; the Upper Cretaceous pelagic "Senonian limestones" with microbreccia and calciturbidites, and with nodular and siliceous beds within the calciturbidites; the Paleocene "limestones with microbreccia"; the Eocene "platy limestones"; and the Middle Eocene to Middle Miocene Deep sea fans (Figure 2b). The Jurassic to Eocene deposits present thickness variations from 0 m to 300 m, for each different age formation, across the basin, due to the half-graben geometry of the independent sub-basins [17,28].

According to [27–29], the Lower Cretaceous "Vigla" limestones in the external Ionian sub-basin consist of white, light-grey-to-yellowish micrites and radiolarian biomicrites, usu-ally thin-bedded to platy with chert intercaltions and chert nodules. The middle sub-basin includes yellow-to-red marly limestones or shaly limestones and chert alternations, as well as clay layers that are usually green and red. The calcareous beds consist of micrites, biomi-crites with foraminifera and radiolaria, and siliceous biomicrites (Si-wackestones, pack-stones). The internal sub-basin comprises compact, thick-bedded, bitumenious, dolomitic limestones, with lenses of slightly dolomitized microbreccia and thin cherty layers, chert intercalations, and nodules. Dolomitization is restricted to rocks of Barremian-Aptian age.

The Upper Cretaceous Senonian limestones, described as "clastic limestones" in the external sub-basin, consist of floatstones, rudstones, and rare grainstones and packstones with a micritic matrix, and micritic or biomicritic intercalations. Occasionally, chert nodules and yellowish layers are also observed. In the middle sub-basin the limestones are characterized as microclastic, bioclastic, or microbreccias with a micritic matrix, intercalated with micrites and biomicrites. Chert layers are rarely observed. In the internal sub-basin limestones are massive, with thick bedded microbreccias to breccias containing rudists and coral fragments. Rarely, layers of platy to thick-bedded micritic to biomicritic limestones are intercalated, whereas nodules and thin layers of chert are observed locally.

The Paleocene rocks, termed as "limestones with microbreccia" formation, are characterized similarly to the Upper Cretaceous (Senonian) limestones lithofacies. These rocks, with prominent microbreccia, are derived from the erosion of Cretaceous carbonates from both the Gavrovo platform (to the east) and the Apulian platform (to the west). At some horizons, bedded or nodular chert is also found.

The Lower Eocene rocks comprise "platy limestones" and platy wackestone/mudstone with Globigerinidae and nodular chert, especially in the central area of the Ionian basin. No siliceous beds were identified in the Lower Eocene "platy limestones" formation.

## 3. Methods

The present work was conducted on the basis of detailed field studies of siliceous concretions (nodules) and siliceous beds. Additionally, stylolites and secondary fractures were measured on the hosted rocks or around them. Measurements took place in the Cretaceous and Palaeocene limestones, and outcropping on Kefalonia island (Figure 3a), in the APM (Pre-Apulian zone), and on Ithaca, Atokos, and Kastos islands, in the external Ionian sub-basin (Figure 3b).

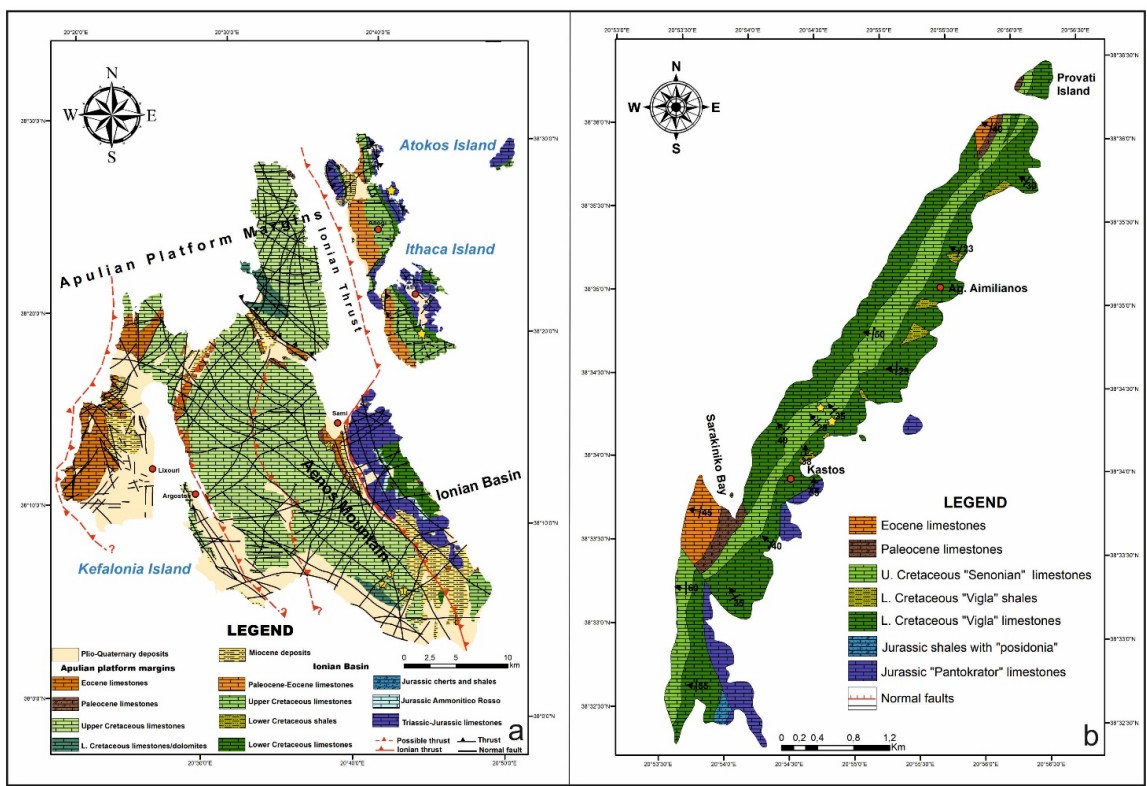

**Figure 3.** Geological maps of (**a**) Kefalonia, Ithaca, and Atokos islands and (**b**) Kastos island [1,29].

Fifty-six samples from thirty nodules specimens were analyzed using XRPD, selected from eleven different sampling sites. From the aforementioned samples, thirty-five samples

are representative of the APM and the remaining twenty-one of the IB. Our results are correlated with thirty-nine pre-existing analyzed samples from Kastos Island that correspond to twelve nodules and sixteen siliceous bed specimens [1]. Our study aims to interpret the depositional conditions that prevailed internally in the IB, as well as to compare them with related results regarding the APM. Most of the nodules were cut along their long diameter in order to analyze the different concentric parts presenting different coloration, whereas in siliceous beds the different color lithologies were sampled separately in the field.

Selected samples were ground into fine powder in either the total sample or independently for each different zone and were analyzed using X-ray powder diffraction (XRPD). Most of the nodules that were sampled separately in the field were cut along their along diameter in order to analyze the different concentric lithologies. Samples were ground (<10 μm) in a vibration disc mill using an agate grinding set and randomly mounted in a sample holder. The XRPD data were collected at the Minerals and Rocks Research Laboratory, Department of Geology, University of Patras, under a Bruker D8 Advanced Diffractometer, using Ni filtered Cu-Kα radiation, operating at 40 kV and 40 mA and employing a Bruker Lynx Eye fast detector. Samples were step-scanned from 2° to 70° with a step size of 0.015° (2θ). For the identification of crystalline phases, the DIFFRACplus EVA (Bruker-AXS, Madison, WI, USA) software was used, based on the ICDD Powder.

Representative-oriented limestone rock specimens were collected from each studied locality and were thin sectioned for microfacies and biostratigraphic analyses, in order to provide a determination of the depositional palaeoenvironments and the age of deposition, respectively.

## 4. Description of the Studied Outcrops and Their Stratigraphic and Sedimentological Setting

Outcrops from three different areas, i.e., Kefalonia island (Apulian platform margins), Ithaca, and Atokos islands (external Ionian zone), were studied (Figure 3a), and all the collected data were correlated with previous research from Kastos island [1] (Figure 3b), as well as part of the external Ionian sub-basin (Table S1).

### 4.1. APM: Kefalonia Island

Upper Cretaceous deposits, according to the pre-existing geological map, mostly characterize Kefalonia island, whereas Lower Cretaceous deposits have outcrops only in two restricted areas (Figure 3a). Specimens were collected from different areas, and they seemed to belong to the Lower Cretaceous carbonates (Figure 3a). Detailed sampling took place at ten sites, suggesting four different sample groups, based on their age determination (Table 1).

Specimens of groups 1 and 2, from outcrops in the western side of Aenos Mountain, are characterized by strong diagenetic processes and were accumulated during the Early Cretaceous. Group 3 specimens came from outcrops in the central part of Aenos Mountain, east of the location of the previous groups, and these rocks were accumulated during the Upper Cretaceous, whereas group 4 specimens correspond to Palaeocene age and were collected from the eastern side of Aenos Mountain, at Sami area (Figure 3a, Table 1).

In groups of Cretaceous age (1–3), mostly small spherical concretions were accumulated (5–10 cm in diameter), with some exceptions reaching up to 25–30 cm in diameter (Figures 4–7). Concretions that developed along the bedding planes were also observed, which were up to 50–60 cm long and 3–5 cm thick. In the respective Paleocene rocks, the group (4) siliceous concretions are relatively large; their length often exceeds 100 cm, and their thickness is usually greater than 30 cm (e.g., Figures 8 and 9). All concretions developed within thin-to-medium-bedded carbonate successions.

Outcrops study shows that fluid flows took place either along the bedding surfaces, producing "boudinage"-like concretions or through stylolites, in places where concentrations of nodules were formed (Figures 5, 6 and 8). These carbonates rocks are characterized by a dense network of stylolites (e.g., Figures 6, 8, 9 and 10a,b). Moreover, the formation of concretions produced secondary fractures, and more specifically both in the overlying and underlying beds, in places where they were thickest (e.g., Figures 5, 6 and 8b).

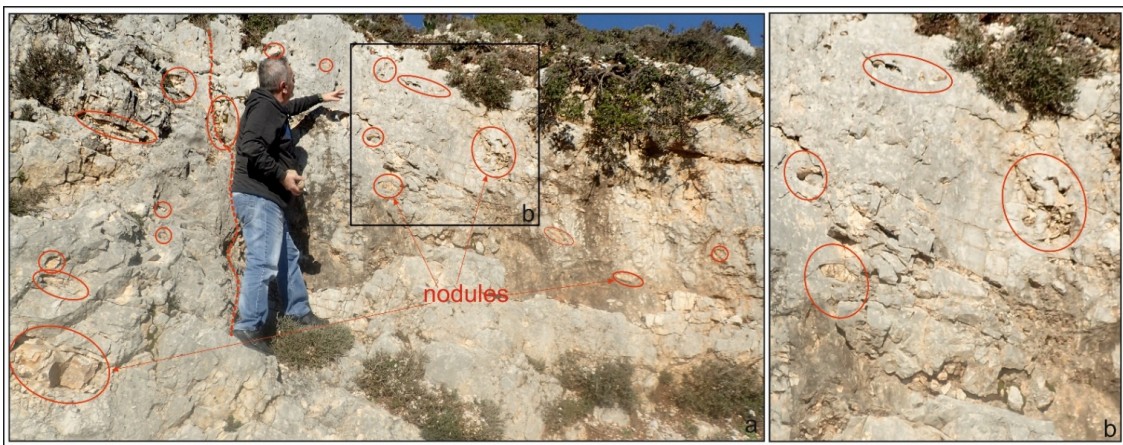

**Figure 4.** Outcrop of thin to medium-bedded limestones, with abundant, small-sized nodules and few bedded siliceous concretions. (**a**) a panoramic view of the outcrop. Black box marks figure, (**b**) Different size siliceous concretions are shown within the red cycles.

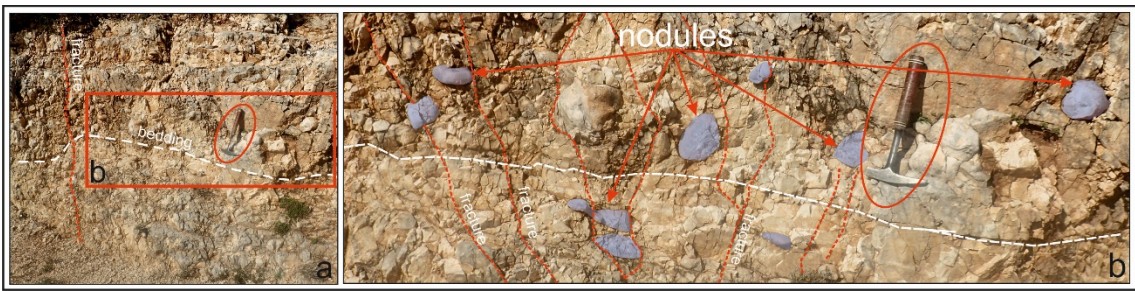

**Figure 5.** (**a**) Panoramic view of the studied section; (**b**) abundant secondary fractures (red lines), related to the presence of siliceous concretions (with greyish tints), are highlighted. For their exact location, see Figure 5a. Dashed white lines in both figures show the bedding plane.

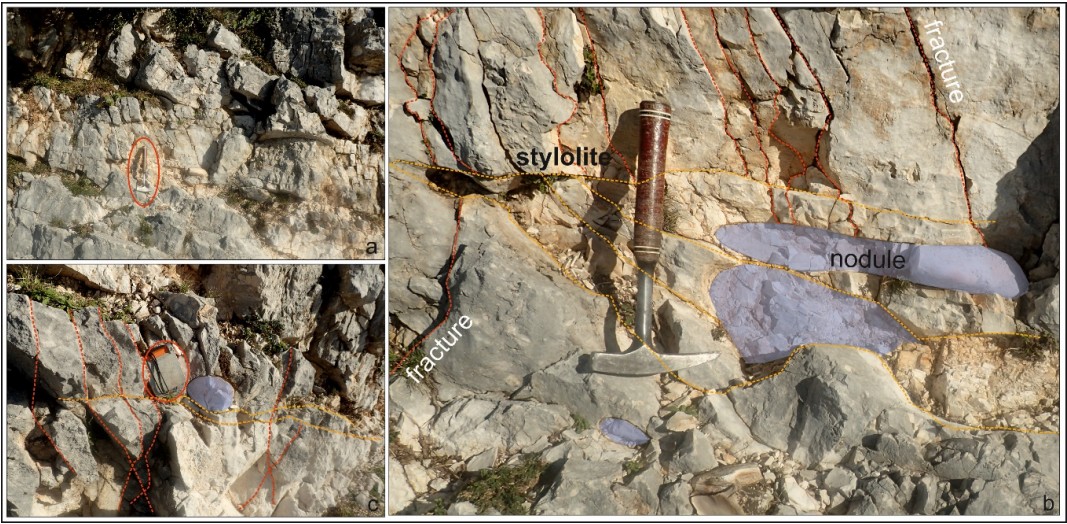

**Figure 6.** Two different sampling sites showing the same siliceous concretions development on the bedding planes. Bedding planes were used for fluid flows as stylolites (orange dashed lines). The red lines show the secondary fractures formed vertically to the concretions long axis. (**a**) Both bedded and siliceous concretions are shown within thin to medium bedded limestones, (**b**) Characteristic stylolites that supported the development of siliceous concretions, (**c**) A characteristic view of how secondary fractures cut only some of the surrounding limestones.

**Table 1.** Representative selected and analyzed samples with the description of microfacies analysis and their respective age determination, for Kastos, Ithaca, and Kefalonia islands.

| A/A | S.N. | Thin Section Facies Analysis | Fossils | Age Based on Fossils |
|---|---|---|---|---|
| | | **Kastos** | | |
| 1 | KI76 | Mudstone. SMF3/FZ1 | Radiolaria, *Rotalipora cushmani*, *Clavihedbergella simplex*, and *Hedbergella planispira* | Upper Cretaceous (Cenomanian) |
| 2 | KI72 | Wackestone/packstone, microbrecciated, and skeletan grains. SMF4/FZ3 | Miliolidae, Bivalve fragments, and *Parasubbotina pseudobulloides* | Upper Cretaceous (Maastrichtian) |
| 3 | KI70 | Wackestone, skeletal graines, and micrites. SMF3/FZ1 | Radiolaria, *Clavihedbrgella simplex*, *Hedbergella planispira*, *Rotalipora cushmani*, *Praeglobotruncana delrioensis*, *Thalmanninella appenninica*, *Thalmanninella greenhornensis*, *Thalmanninella globotruncanoides*, and *Whiteinella archaeocretacea* | Upper Cretaceous (Cenomanian) |
| 4 | KI68 | Packstone, microbioclastic, and microcrystalline. SMF2/FZ1-2 | *Blefuscuiana gorbachikae*, *Blefuscuiana occulta*, *Blefuscuiana praetrocoidea*, *Globigerinelloides* cf. *ferreolensis*, *Globigerinelloides gottisi*, *Globigerinelloides barri*, and *Globigerinelloides blowi* | Lower Cretaceous (Aptian) |
| 5 | KI67 | Wackestone, microbioclastic. SMF2/FZ1 | Radiolaria, *Planomalina cheniurensis*, *Globigerinelloides* sp., *Blowiella blowi*, *Blowiella gottisi*, and *Hedbergella* sp. | Lower Cretaceous (Aptian) |
| 6 | KI60 | A. mudstone with micrite skeletal grains B. microbioclastic, packstone, microcrystalline, and lamination. SMF3/FZ1 B. SMF2/FZ1-2 | Radiolaria, *Hedbergella rischi* | Lower Cretaceous (Albian) |
| | | **Ithaca-Gidaki** | | |
| 7 | Gidaki | Bioclastic floatstone/rudstone with reef-derived material. Inter-and intraparticle blocky cement. Intense meteoric diagenesis. SMF6/FZ4 | *Orbitolina* sp., Rudist fragments, Echinoderma fragments, and *Cuneolina* sp. | Lower to Upper Cretaceous (Albian to Cenomanian) |
| 8 | Gidaki | Bioclastic rudstone with reef-derived material. It is characterized by high porosity, due to extensive meteoric shell solution. Biomolds have been filled, partially or entirely, by blocky calcite cement. Most of molds are related to pedogenesis, due to extensive subaerial exposure. Abundant coarse bioclasts of rudists and, in less proportion, echinoderms. Intense meteoric diagenesis and pedogenesis. Most biomolds remained uncemented or cemented partially, with the rest of the space remaining open. In molds or cavities, remains of organic material have been observed. SMF6/FZ4 | *Orbitolina* sp., Rudist fragments, and Echinoderma fragments | Lower to Upper Cretaceous (Albian to Cenomanian) |
| 9 | Cave Rizes | Calciturbidite (alternation of calcarenite-calcirudite). Bioclasts of rudists and echinoderms transported in pelagic environment. Intense meteoric diagenesis (micritization). SMF4/FZ4 | Radiolaria, *Ticinella* sp. | Lower Cretaceous (Albian) |

**Table 1.** *Cont.*

| A/A | S.N. | Thin Section Facies Analysis | Fossils | Age Based on Fossils |
|---|---|---|---|---|
| | | **Kefalonia** | | |
| 10 | group 1 Aenos central | Mudstone/wackstone, biomicrite with a few non skeletal clasts. Stylolites can be observed. SMF3/FZ3 | Radiolaria (Spumellaria), *Hedbergella trocoidea*, *Clavihedbergella simplex*, and *Blefuscuiana gorbachikae* | Lower Cretaceous (Lower Albian) |
| 11 | group 2 Aenos west | Wackestone with a few scattered ooids, recrystalised locally or sometimes more extensively with well-formed dolomite crystals, dolomitic limestone, or dolomite; locally bitoumenous. SMF3/?FZ1 | Radiolaria, filaments | Lower Cretaceous |
| 12 | // | Crystalline, recrystalised locally, or sometimes more extensively (dolomitized) with well-formed dolomite crystals, dolomitic limestone, or dolomite; bitoumenous. SMF3/?FZ1 | Radiolaria? | Lower Cretaceous |
| 13 | // | Wackestone, locally dolomitised with well-formed dolomite crystals; bitoumenous; stylolites are observed. SMF3/FZ1 | Radiolaria, Calpionellidae | Lower Cretaceous (Tithonian-Valanginian) |
| 14 | group 3 Aenos central | A. Bioclastic packstone to rudstone with ooids; large bioclast fragments can be observed. SMF5/FZ4 B. Wackestone/packstone with a few planktonic foraminifera. SMF3-4/FZ3 | Radiolaria, Algae fragments, Rudist fragments, Mollusk fragments, *Cuneolina* sp., *Orbitolina* sp., and *Hedbrgella* sp. | Lower to Upper Cretaceous (Late Albian–Late Cenomanian) |
| 15 | // | Microbrecciated limestone with ooids, exoclasts with peloids, biomicrite, and wackestone/packstone. SMF4-5/FZ4 | Radiolaria (Spumellaria), Nasselaria, Mollusc fragments, Algae, Miliolidae, *Orbitolina* sp., *Hedbergella trocoidea*, *Hedbergella planispira*, *Globigerinelloides* sp., and *Biglobigerinella barri* | Lower Cretaceous (Late Alptian–Early Albian) |
| 16 | // | A. Crystalline, recrystalised extensively (dolomitised) with well-formed dolomite crystals, dolomitic limestone, or dolomite; bitoumenous. B. Packstone to rudstone; pelloids are present. SMF5/FZ4 | Algae, Milliolidae, and Mollusc fragments | Lower Cretaceous |
| 17 | // | Packstone with scattered ooids, parts with pelloids in a sparitic matrix (exoclasts?), a few fenestral cavities, and large reef fragments. FZ4/SMF5 | Algae, Plaktonic foraminifera | |
| 18 | group 4 Sami | Wackestone, microbioclastic. SMF3/FZ3 | Radiolaria, *Morozovella angulata*, *Morozovella aequa*, *Morozovella conicotruncana*, *Morozovella velascoensis*, *Morozovella occlusa*, *Planorotalites chapmani*, *Subbotina velascoensis*, *Subbotina triangularis*, *Globanomalina ehrenbergi*, *Igorina* sp., and *Acarinina* sp. | Paleocene (Selandian–Thanetian) |

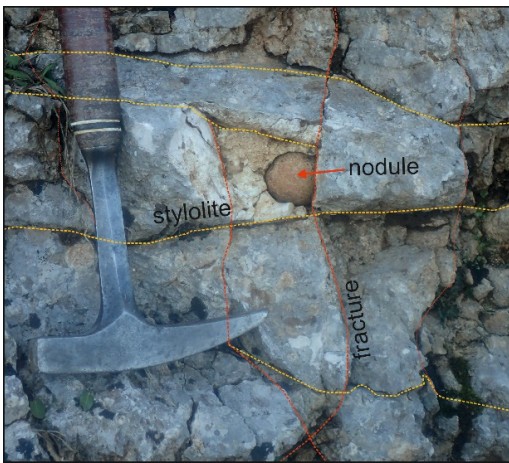

**Figure 7.** A small spherical siliceous concretion internally to the rock. Orange dashed lines indicate the stylolites, whereas the red lines show the fractures.

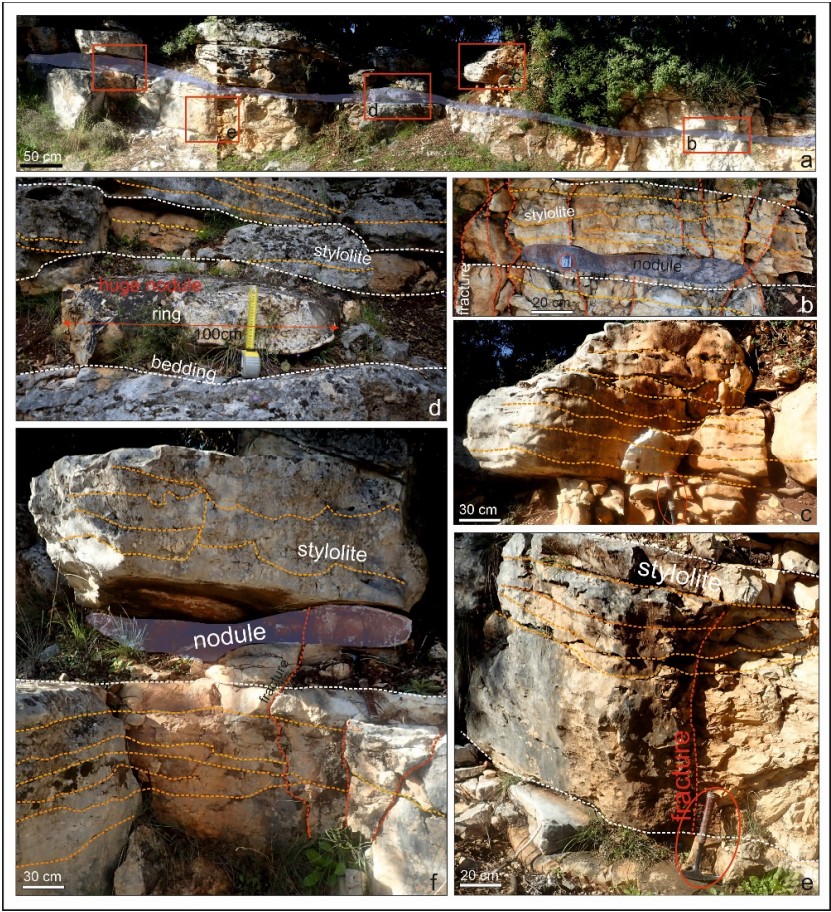

**Figure 8.** (**a**) Panoramic view of a Paleocene outcrop, with siliceous concretions, with five zoom-in photos; (**b**) The siliceous elongated nodule (with greyish shadow) and the abundant secondary fractures (red dashed lines) are marked; (**c**) Part of a rock with several stylolite planes (dashed orange lines); (**d**) Within the major nodule horizon, a nodule more than 100 cm long and 30 cm thick is marked (used also as a scale). Dashed white lines show the bedding planes; (**e**) As in Figure 8b, stylolites, the bedding planes (white dashed lines), and some fractures are marked; (**f**) The boudinage-like, in-shape, siliceous nodule is marked, and many stylolites can be seen in the overlying and underlying horizons.

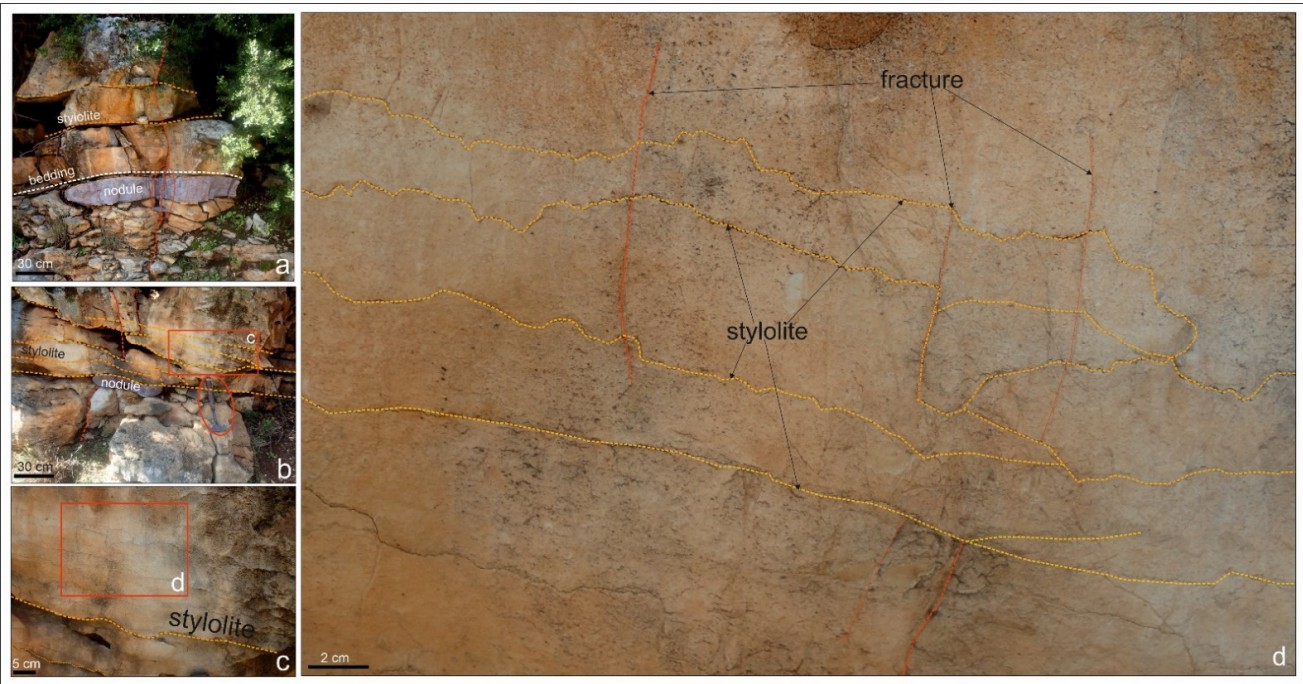

**Figure 9.** Bedding planes are marked in all photos with white dashed lines, fractures with red dashed lines, and stylolites with orange dashed lines. (**a**) Huge nodule within the bedding plane; (**b**) abundant stylolites inside the beds related with the nodules development. Red box shows the position of Figure 9c; (**c**) From close the stylolilites within the beds; (**d**) shows in detail the above-described fractures and stylolites from Figure 9b,c.

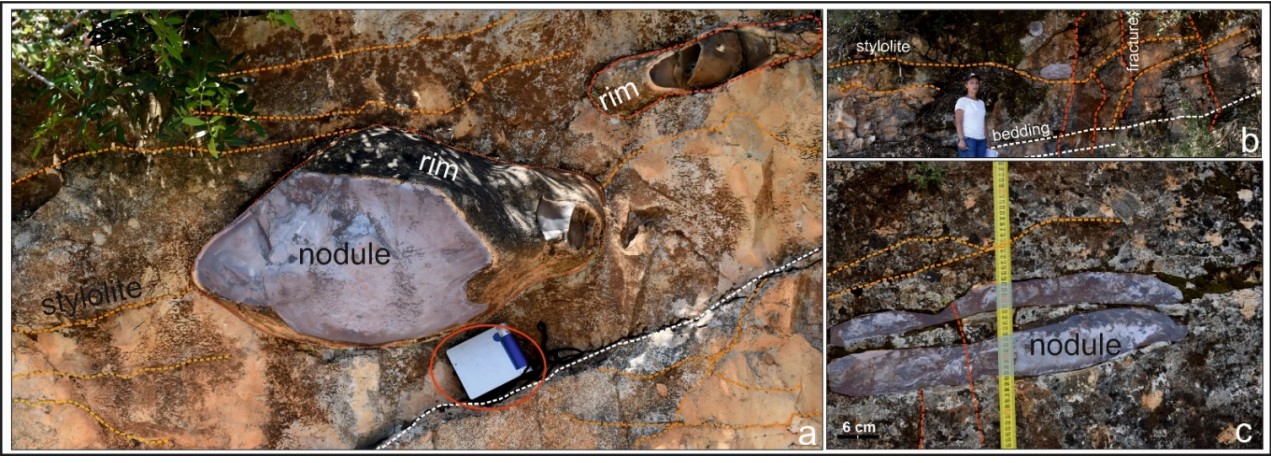

**Figure 10.** Nodules of Paleocene age collected from the same Sami area. (**a**) Note the huge nodule with the characteristic rim and the stylolites around nodules. For scale, use the compass; (**b**) a characteristic image showing bedding planes, stylolites and secondary fractures around nodules. The total high of Dr. Zoumpouli is 160 cm; (**c**) Secondary fractures that cut the nodules and underlying limestones but not the overlying limestones. Nodules developed internally to the bedding planes forming lenticular geometries. The vertical scale is 55 cm.

### 4.2. IB: Ithaca and Kastos Islands

In Ithaca (Figures 11 and 12) and Atokos islands, the siliceous concretions were formed within the Lower Cretaceous deposits and developed internally in thin to medium-bedded limestones, especially within the bedding planes (Figure 12b), and are also present within deformed horizons (Figure 12c). Moreover, secondary fractures were produced due to the development of these concretions (Figure 12a).

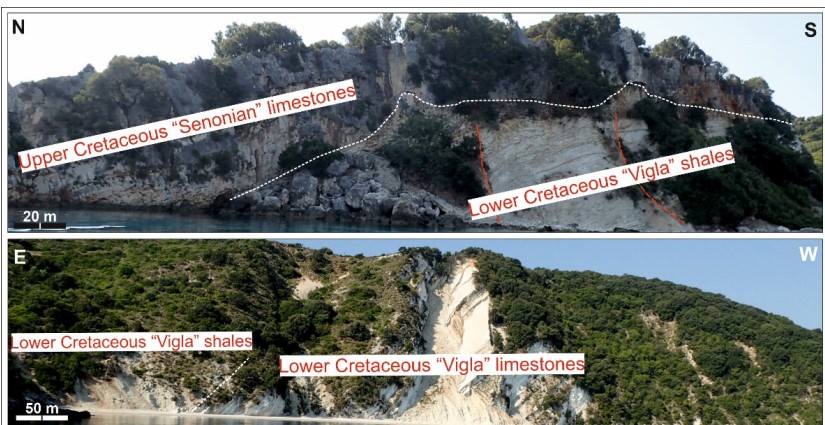

**Figure 11.** Panoramic view of Gidaki beach in Ithaca island. In these photos, the studied succession is presented. Lower Cretaceous deposits "Vigla" Formation are situated beneath the Upper Cretaceous "Senonian" limestones.

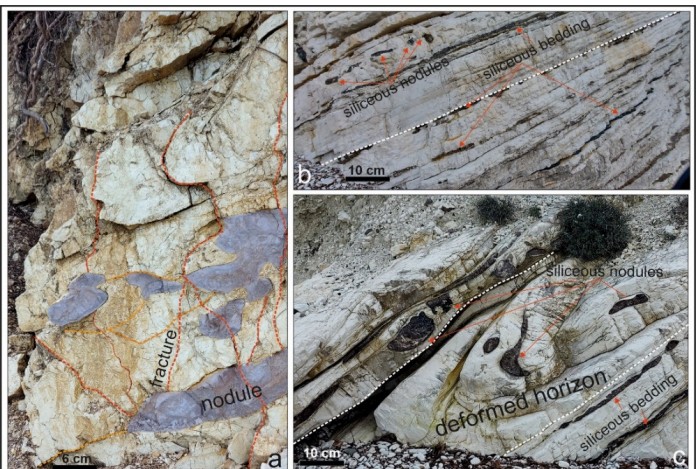

**Figure 12.** (**a**) Siliceous concretions within the medium-bedded limestones where many secondary fractures (red dashed lines) occur; (**b**) both siliceous nodules and siliceous beds were found; and (**c**) siliceous nodules within a slumped horizon. White dashed lines show the bedding planes.

Specimens from Kastos island have been described in a previous work [1] and were employed in the present study regarding only their mineralogical correlation. Lower Cretaceous "Vigla" limestones contain a variety of nodules ranging in diameter, from 1 to 15 cm, as well as in color, i.e., white, grey, and dark grey. The siliceous nodules from the Upper Cretaceous "Senonian" limestones can reach 20 cm in diameter and occur between sedimentary cycles and close to the top of structureless calcarenite beds.

## 5. Description of Selected Concretions in the Outcrops and in the Laboratory after Cutting Them

Independently of the sampling area, there are concretions with or without any difference from core to the ring. Most of the specimens are spherical or even they have a more elongated shape. In detail:

### 5.1. Kefalonia Island

Group 1 includes specimens from Aenos mountain (west and central Cretaceous age) presenting either a general homogeneity or differentiating from core to rim. In this group, the core of the nodule is black in color (Figures 13 and 14). On the other hand, cores from specimens of group 2 have greyish color (Figure 15), whereas in group 3 the studied

specimens do not show any internal differentiation (Figure 16). Paleocene concretions, east Aenos, in Sami area, showed the presence of cores and thick enough white rims. Core color ranges from greyish to bright or dark brown (Figure 17).

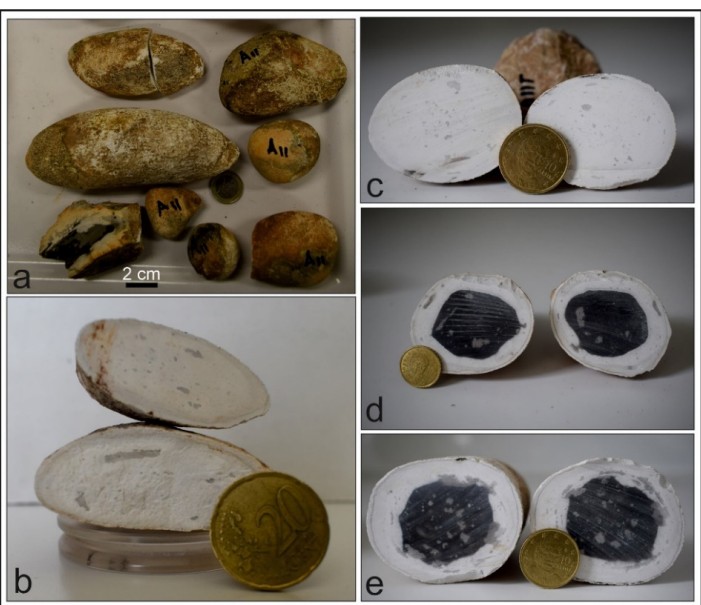

**Figure 13.** (**a**) Representative nodules of different size and shape from group 1 sample A11. The coin of one euro is for scale and is 2.3 cm in diameter; (**b**,**c**) without any significant internal difference. For scale the coin of 20 cents is 2.2 cm and the coin of 10 cents is 2.0 cm, and (**d**,**e**) with a characteristic black core and thick white rim. The coin is 2.0 cm.

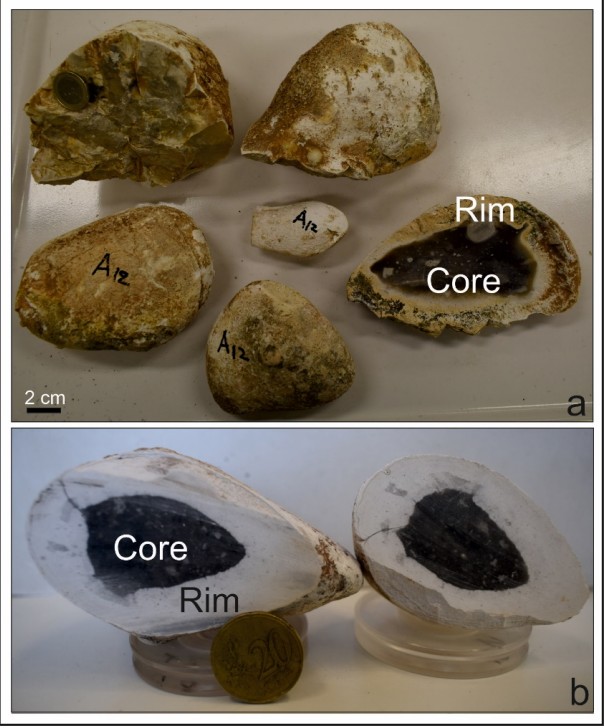

**Figure 14.** Representative nodules of different sizes and shapes from group 1 sample A12 (**a**,**b**) show the thick black core and the thick white rim across the nodules' cutting. The coins for scale are: one euro 2.3 cm and 20 cents 2.2 cm, respectively.

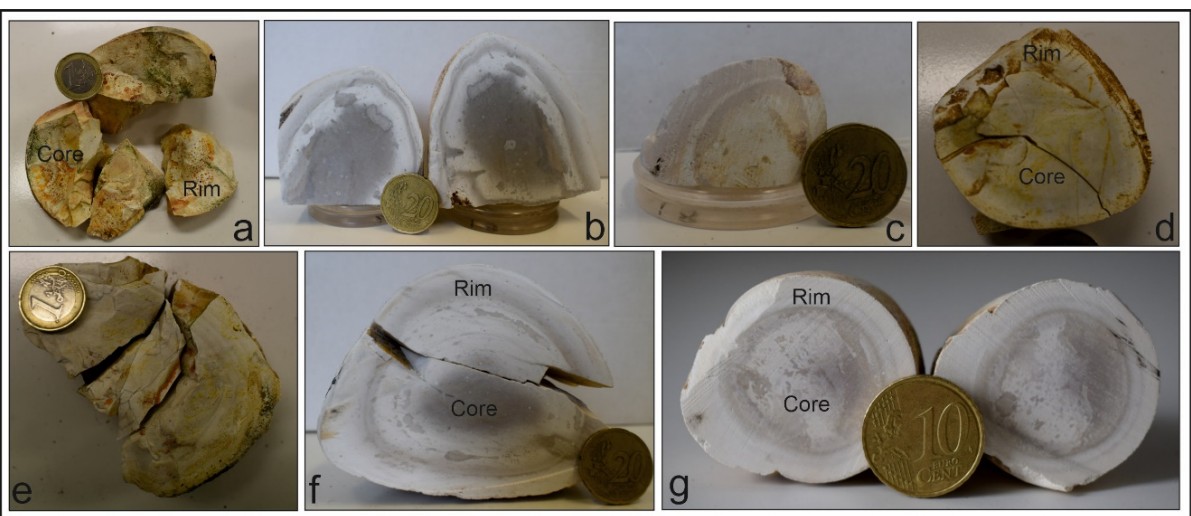

**Figure 15.** Nodules of Lower Cretaceous deposits belonging to group 2. (**a**) sample A2, (**b**) sample A3, (**c**) sample A4, (**d**) sample A5, (**e**) sample A6, (**f**) sample A7, and (**g**) sample A8, where the core is lighter in color than the previous studied samples, with a white-to-greyish color and not a dark one. The coins for scale are: one euro 2.3 cm, 20 cents 2.2 cm, and 10 cents 2.0 cm, respectively.

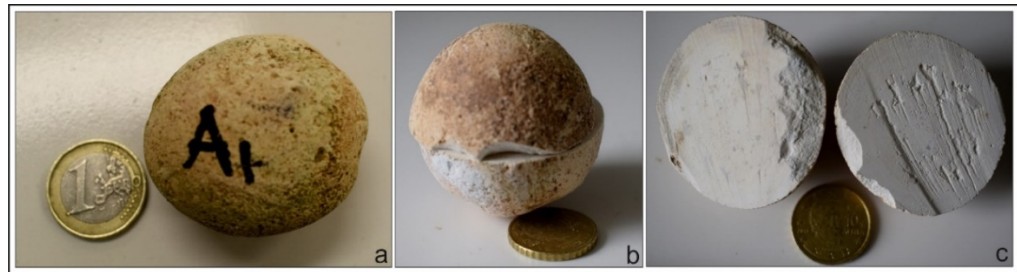

**Figure 16.** Nodules of Upper Cretaceous deposits from group 3 sample A1. (**a**) before cut it; (**b**) after we cut it; (**c**) internally without any significant color difference from core to rim. Only small parts of the rim seem to present a darker horizon.

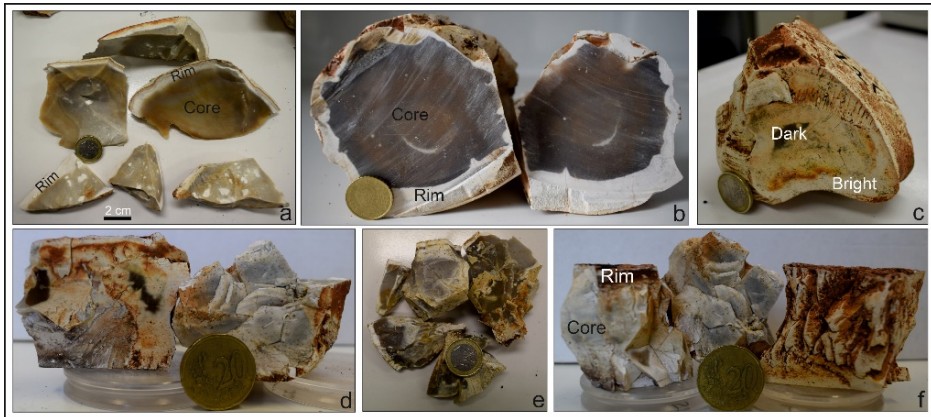

**Figure 17.** Samples of Paleocene age from Sami area, group 4. (**a**) Sample S1, (**b**) sample S1b, (**c**) sample S2c, (**d**) sample S3a, (**e**) sample 4a, and (**f**) sample S4b. The color of the core is quite different than that of the previous presented specimens. The coins for scale are: one euro 2.3 cm, 20 cents 2.2 cm, and 10 cents 2.0 cm, respectively.

*5.2. Ithaca and Atokos Islands*

In Ithaca island, the studied nodules that were collected from two different areas, the southern (Rizes Cave) and central (Gidaki beach), showed many internal differences. Samples from Gidaki beach present many differences from the one specimen to the other (Figure 11). There are nodules with a black core and nodules with white or bright brownish cores or either with two different core colors (Figure 18). Specimens from southern Ithaca island are mostly spherical with a greyish color core, with one or two cores and rims presenting spherical color gradations (Figure 19).

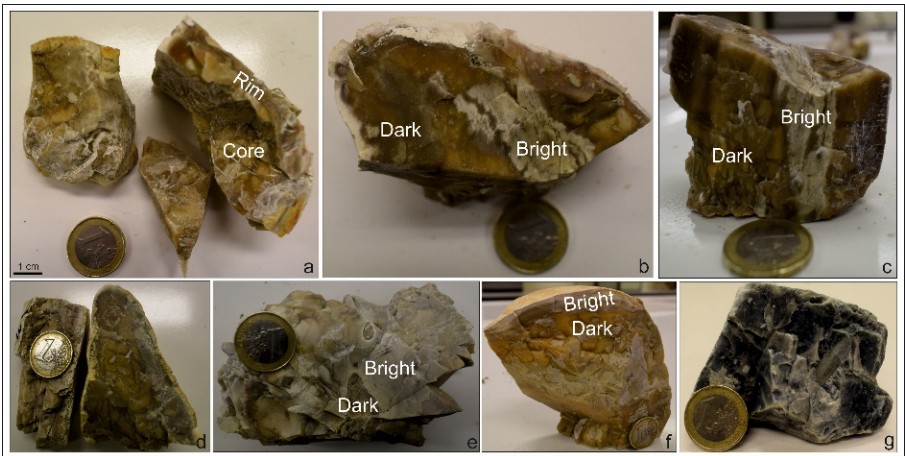

**Figure 18.** Siliceous nodules from Gidaki beach in central Ithaca Island Lower Cretaceous in age, with black core, with white or bright brownish cores or either with two different core colors. (**a**) sample G1, (**b**,**c**) sample G2, (**d**) sample G3, (**e**) sample G4, (**f**) sample G5, and (**g**) sample G6. The coin for scale of one euro is 2.3 cm.

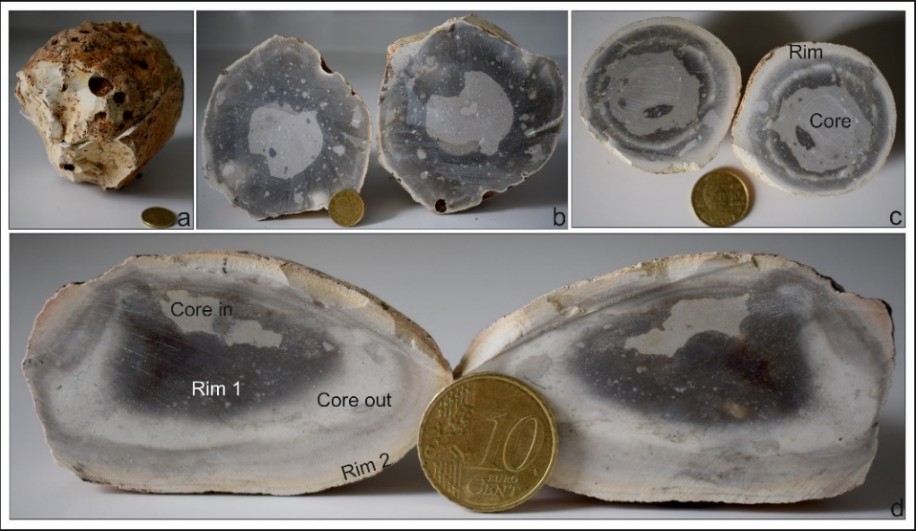

**Figure 19.** Samples from southern Ithaca island (Rizes Cave) are mostly spherical with greyish color core, with one or two cores and rims (Figure 18d) presenting spherical color graduations. (**a**,**b**) sample ITHS1, (**c**) sample ITHS3, and (**d**) ITHS2. The coins for scale is of 10 cents and it is 2.0 cm.

Specimen from Atokos island, compared to all other specimens, shows a remarkable difference, in which the core is white and the rim is black (Figure 20).

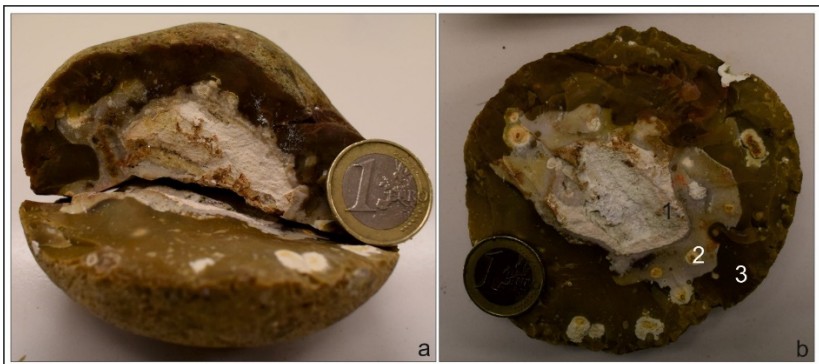

**Figure 20.** Sample from Atokos island show a significant difference compared to all other samples, with a white core and a dark rim. (**a**) Shows the geometry of the total nodule. The lower part represents the half nodule; whereas the upper part only the $\frac{1}{4}$ of the nodule, giving a 3D image of the nodule; (**b**) the same half nodule showing the internal organization into three horizons. The coin is 2.3 cm.

*5.3. Kastos Island*

The Lower and Upper Cretaceous nodules differ in color from the outer to the middle part, from lighter to darker colors. Furthermore, nodules consist mostly of one to two, or rarely even three, different parts with spherical shapes, forming the core, the main body, and the rim, having distinguishable color differences between the different parts. The nodules differ in color having white, red-brown, or greyish rims and greenish cores, and the diameter of the nodules ranges from 1 to 30 cm.

## 6. Mineralogical Analysis by Means of XRPD

The XRPD analysis of the powder samples from Kastos island (Table 2), from a previous work [1], showed that nodules consist of two major minerals (quartz and calcite) and two minors ones (moganite and dolomite). The presence of moganite and Opal-A suggests an amorphous silica precursor. The presence of dolomite in the analyzed samples can be related to the underlying Early Cretaceous dolomites, whereas the presence of maghemite is related to the presence of ferrous oxides in the flowing water solutions. The different mineralogical compositions of the nodules arranged in concentric spherical zones suggest the selective replacement of calcite with silica. The fact that nodules are not flattened but spherical supports the idea that they developed largely diagenetically after the sedimentation and deformation of the hosted deposits.

**Table 2.** Mineralogical analysis of forty-eight powder samples from thirty-nine samples of chert. Nodules from different levels were analyzed independently for each level. Table is organized into a. different age Early Cretaceous ("Vigla" limestones and "Vigla" shales) and Late Cretaceous (Senonian limestones), and b. different source (nodules or siliceous beds). The minerals present include quartz, moganite, calcite, and dolomite, as well as others and an unknown reflectance. With red and blue colors, the same samples but from different areas within the nodules (core, main body, or rim) are marked.

| Sample | S/N | Position | Quartz | Moganite | Calcite | Dolomite | Others |
|--------|-----|----------|--------|----------|---------|----------|--------|
| Kastos Island-Lower Cretaceous Vigla Formation | | | | | | | |
| C1. Nodules | | | | | | | |
| VK1 | 40 | main body | +++++ | + | - | - | |
| VK2 | 41 | main body | +++++ | + | +/++ | - | opal-A |
| VK3 | 42 | main body | +++++ | + | +++ | - | |
| VK4a | 43a | rim | + | - | +++++ | tr | |
| VK4b | 43b | main body | +++++ | + | tr | - | opal-A |

**Table 2.** *Cont.*

| Sample | S/N | Position | Quartz | Moganite | Calcite | Dolomite | Others |
|---|---|---|---|---|---|---|---|
| | | | **C2. Bedded cherts** | | | | |
| VP1 | 44 | main body | +++++ | + | tr | - | |
| VP2 | 45 | main body | +++++ | + | tr | tr | |
| VP3 | 46 | main body | +++++ | + | - | tr | maghemite |
| VP4 | 47 | main body | +++++ | + | - | tr | maghemite, opal-A |
| VP5 | 48 | main body | +++++ | + | - | tr/+ | maghemite, opal-A |
| VP6 | 49 | main body | +++++ | + | - | tr/+ | opal-A |
| VP7 | 50 | main body | +++++ | + | ++++ | - | maghemite |
| VP8 | 51 | main body | +++++ | - | - | tr | maghemite |
| VP9 | 52 | main body | +++++ | tr | - | tr | maghemite |
| | | | **Kastos Island-Upper Cretaceous Senonian Formation** | | | | |
| | | | **D1. Nodules** | | | | |
| SK1 | 53 | main body | +++++ | + | ++ | - | opal-A, maghemite |
| SK2 | 54 | main body | +++++ | + | ++ | - | |
| SK3a | 55a | rim | ++++ | - | +++++ | - | |
| SK3b | 55b | main body | +++++ | tr | ++ | - | |
| SK4a | 56a | main body | +++++ | + | ++ | - | Mg-cal |
| SK4b | 56b | core | ++++ | - | +++++ | - | |
| SK4c | 56c | dark grey dikes | ++++ | - | ++ | - | opal-A |
| SK5a | 57a | core | +++ | - | +++++ | - | |
| SK5b | 57b | main body | +++++ | + | ++ | - | opal-A |
| SK5c | 57c | rim | +++++ | + | ++ | - | |
| SK6a | 58a | rim | +++++ | + | +/++ | - | |
| SK6b | 58b | main body | +++++ | + | + | - | |
| SK7a | 59a | rim | +++++ | - | ++++ | - | |
| SK7b | 59b | main body | +++++ | + | tr | - | maghemite |
| SK8 | 60 | main body | +++++ | + | tr | - | maghemite |
| | | | **Bedded Cherts** | | | | |
| SP1 | 61 | main body | +++++ | tr/+ | ++ | - | |
| SP2 | 62 | main body | +++++ | tr/+ | ++ | - | opal-A |
| SP3 | 63 | main body | +++++ | tr/+ | tr | - | |
| SP4a | 64a | dark red main body | +++++ | tr | - | - | maghemite |
| SP4b | 64b | light red main body | +++++ | tr/+ | tr | - | maghemite |
| SP5 | 65 | main body | +++++ | tr | ++++ | - | |
| SP6a | 66a | grey main body | +++++ | - | + | - | |
| SP6b | 66b | white main body | +++ | - | +++++ | - | |
| SP6c | 66c | grey-red main body | +++++ | tr | +++ | - | |
| SP7 | 67 | main body | +++++ | tr/+ | +/++ | - | |

-: not detected; tr: traces; +: few; ++: common; +++: frequent; ++++: abundant; and +++++: dominant.

The XRPD analysis, in Kefalonia island specimens (situated in the APM) (Table 3) and in Ithaca and Atokos islands specimens (representing the western boundaries of the external Ionian sub-basin) (Table 4), showed one major mineral (quartz), one mineral with a less major impact (moganite), and one minor mineral (calcite) (Figure 21a,b).

**Table 3.** Mineralogical analysis of thirty-four powder samples of siliceous concretions from APM in Kefalonia island. The identified minerals include quartz, moganite, calcite, maghemite, halite, and dolomite. With red and blue colors, the same samples but from different area of nodules (core, main body, or rim) were marked.

| Groups | Sample Code | Sample No | Sampling Area | QZ | MOG | CAL | Maghemite | Hal | Dol |
|---|---|---|---|---|---|---|---|---|---|
| colspan | | | Apulian Platform Margins–Kefalonia Island | | | | | | |
| Group 1 | A11_A | 1 | total | +++++ | ++ | | | | |
| // | A11b_1 | 2a | core | +++++ | tr | | | | |
| // | A11b_2 | 2b | rim | +++++ | + | | | | |
| // | A11d_2 | 3a | core | +++++ | + | | | | |
| // | A11d_1 | 3b | rim | +++++ | ++ | | | | |
| // | A11z | 4 | total | +++++ | ++ | | | | |
| // | A12a_1 | 5a | core | +++++ | + | | | | |
| // | A12A_2 | 5b | rim | +++++ | ++ | | | | |
| // | A12b | 6 | total | +++++ | tr | tr | | | |
| // | A12g_1 | 7a | core | +++++ | + | | | | |
| // | A12g_2 | 7b | rim | +++++ | ++ | tr | | | |
| Group 2 | A2_1 | 8a | core | +++++ | + | | | | |
| // | A2_2 | 8b | rim | +++++ | + | | | | |
| // | A3 | 9 | total | +++++ | + | | | | |
| // | A4 | 10 | total | +++++ | ++ | | | | |
| // | A5_1 | 11a | core | +++++ | + | tr Mg-cal | | | |
| // | A5_2 | 11b | rim | +++++ | + | | | | |
| // | A6 | 12 | total | +++++ | + | | | | |
| // | A7_1 | 13a | core | +++++ | + | | tr < 5% | | |
| // | A7_2 | 13b | rim | +++++ | + | | + from 5 as 15% | | |
| // | A8_1 | 14a | core | +++++ | + | | ++ from 15 as 25% | | |
| // | A8_2 | 14b | rim | +++++ | + | | +++ from 25 as 35% | | |
| Group 3 | A1 | 15 | total | +++++ | ++ | | ++++ from 35 as 50% | | |
| Group 4 | S1_1 | 16a | core | +++++ | +++ | | +++++ more than 50% | | |
| // | S1_2 | 16b | rim | +++++ | ++ | tr | | | |
| // | S1b_1 | 17a | core | +++++ | ++ | tr | | | |
| // | S1b_2 | 17b | rim | +++++ | +++ | | | | |
| // | S2g_1 | 18a | dark | +++++ | +++ | | | | |
| // | S2g_2 | 18b | bright | +++++ | +++ | | | | |
| // | S3 | 19 | total | +++++ | ++ | + | | | |
| // | S4a_1 | 20a | core | +++++ | ++ | tr | | | |
| // | S4a_2 | 20b | rim | +++++ | ++ | tr | | | |
| // | S4b_2 | 21a | core | +++++ | ++ | | | | |
| // | S4b_1 | 21b | rim | +++++ | ++ | | | | |

**Table 4.** Mineralogical analysis of nineteen powder samples of siliceous nodules from Ithaca and Atokos islands, from external Ionian sub-basin. The identified minerals include quartz, moganite, calcite, maghemite, halite, and dolomite. With red and blue colors, the same samples from different areas of the nodules (core, main body, or rim) are marked.

| | | | | | | | | | |
|---|---|---|---|---|---|---|---|---|---|
| colspan | | | **Ionian Basin–Ithaca Island** | | | | | | |
| ITHS1 | 1 | total | +++++ | + | +++ | | | | |
| ITHS2_1 | 2a | core in | +++++ | + | | | tr | tr | tr |
| ITHS2_2 | 2b | rim 1 | +++++ | + | | | tr | | |
| ITHS2_3 | 2c | core out | +++++ | ++ | | | | | |
| ITHS2_4 | 2d | rim 2 | +++++ | + | | | | | |
| G1_2 | 3a | core | +++++ | tr | | | | | |
| G1_1 | 3b | rim | +++++ | + | | | | | |
| G2_2 | 4a | dark | +++++ | + | tr | | | | |

**Table 4.** *Cont.*

| | | | | | | | |
|---|---|---|---|---|---|---|---|
| **Ionian Basin–Ithaca Island** | | | | | | | |
| G2_1 | 4b | bright | +++++ | | tr | | |
| G3 | 5 | total | +++++ | ++ | | | |
| G4_1 | 6a | dark | +++++ | tr | tr | | |
| G4_2 | 6b | bright | +++++ | + | tr | | |
| G4_3 | 6c | more bright | +++++ | + | | | |
| G5_1 | 7a | dark | +++++ | tr | tr | | |
| G5_2 | 7b | bright | +++++ | + | | | |
| G6 | 8 | total | +++++ | + | | tr | tr |
| **Ionian Basin-Atokos Island** | | | | | | | |
| ATOK_1 | 9a | core | ++++ | | +++++ | | |
| ATOK_2 | 9b | rim around the core | + | | +++++ | | |
| ATOK_3 | 9c | rim | +++++ | tr | + | | |

-: not detected; tr: traces; +: few; ++: common; +++: frequent; ++++: abundant; and +++++: dominant.

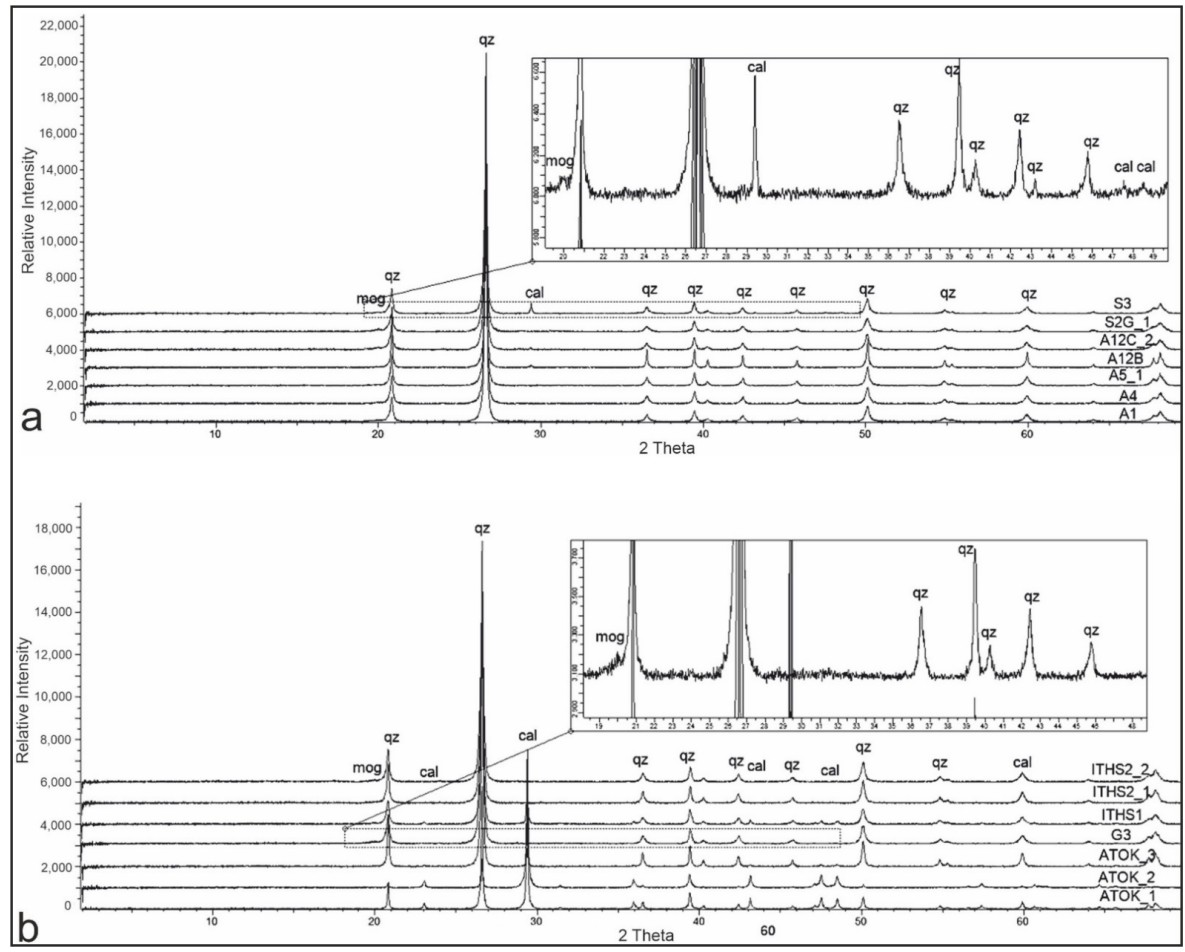

**Figure 21.** (**a**) Seven selected X-ray powder diffractograms for nodules of different age from APM Kefalonia island (for details see Table 3), and (**b**) seven selected X-ray powder diffractograms for nodules from different positions and different age from IB (for details see Table 4). Abbreviation of minerals according to [30]. Selected diffractograms within the same area (both in APM and in IB) show internal differences in moganite, calcite, and quartz content. Inserted boxes represent only a small part of a diffractogram for samples S2G_1 (APM) and G3 (IB), and these selected areas show the moganite reflectance and the differences between the two studied geodynamic settings. On the diffrectograms the symbols represent: Quartz (qz), moganite (mog) and calcite (cal).

In detail:

Quartz is generally the dominant mineral in the concretions of all studied specimens, both in the rim and in the core or in total. Only in one specimen (from Atokos island) did the rim around the core present very low quartz content.

Calcite was absent or in traces in all groups from Kefalonia island specimens but was abundant in the specimen from Atokos island and in some specimens from Gidaki in Ithaca island (mostly in the cores). The rest of the specimens from Ithaca showed only traces of calcite, if present. There was a great difference from Kastos island, where calcite was abundant in the Upper Cretaceous samples and absent in lower Cretaceous siliceous beds.

Moganite was present with significant contents in most studied samples from Kefalonia island, with an increasing content towards the Paleocene specimens from Sami area. In contrast, moganite showed lower quantities in the Ionian Basin specimens, both in Ithaca and Atokos islands. In Kastos island, moganite was present in most samples, with various proportions.

Dolomite, halite, and maghemite were absent from the studied samples of Kefalonia island but present in traces in the Ithaca specimens.

## 7. Cluster Analysis for Results Comparison between Present and Previous Studies

Using cluster analysis (Figure 22) (Table 5), after displacement towards the peak of quartz, correction with Ka2, and taking into account that 1. quantities represent semi-quantitative measurements, and 2. the samples were evaluated according to the height of reflections, the results showed the presence of two main groups:

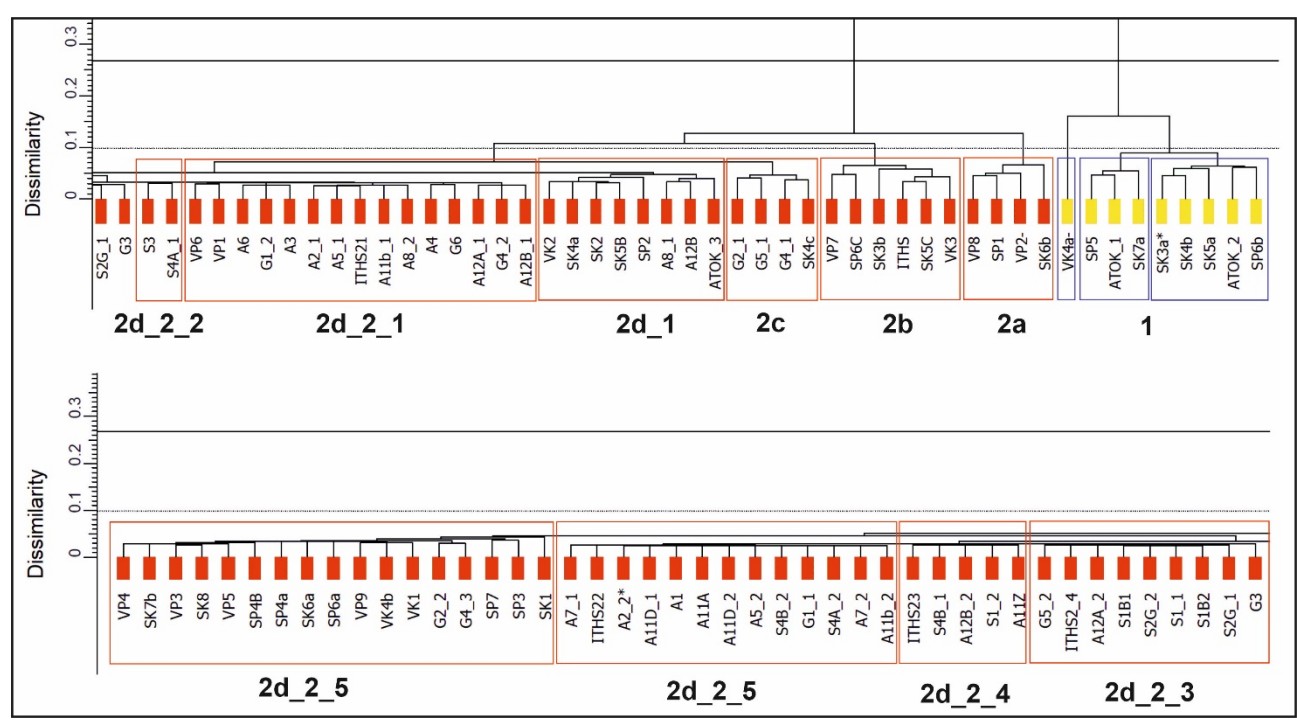

**Figure 22.** Dendrogram obtained from cluster analysis of XRPD patterns for the analyzed samples.

The first group is related to high quantities of calcite, the presence of quartz, and the absence of moganite. These samples belong to the Ionian Basin.

In the second group, moganite is present, and due to the moganite and calcite variations, cluster analysis organizes the samples into many sub-groups. In general, there is a separation in the samples related to the calcite content, showing that higher quantities are present to the Ionian basin, whereas moganite contents are higher in Kefalonia island samples.

**Table 5.** Ranges from the semi-quantitative evaluation of quartz, moganite, and calcite. Samples are grouped according to the cluster analysis of the XRPD patterns.

| Clusters | Quartz (wt %) | Moganite (wt %) | Calcite (wt %) |
|---|---|---|---|
| 1a_1 | 8.6–28.8 | - | 71.2–91.4 |
| 1a_2 | 41.3–66.8 | - | 33.2–58.7 |
| 1b | 2, 5 | - | 97, 5 |
| 2a | 87.1–90.2 | 9.3–11.1 | 0–3.6 |
| 2b | 56.8–89.3 | 0–10.7 | 9.1–39.6 |
| 2c | 94.7–98.1 | 0–3.3 | 1.7–3.1 |
| 2d_1 | 79.9–93.7 | 3.1–8.8 | 0–12.6 |
| 2d_2_1 | 79.9–96.9 | 3.1–19.3 | 0–1.1 |
| 2d_2_2 | 68.8–76.9 | 18.7–20.8 | 4.4–10.4 |
| 2d_2_3 | 73.9–86.3 | 13.7–26.1 | 0 |
| 2d_2_4 | 74.2–85.3 | 16.7–24.2 | 0 |
| 2d_2_5 | 77.8–91.1 | 8.8–21.4 | 0 |
| 2d_3 | 74.4–95.8 | 4.2–20.8 | 0–5.3 |

Even though clustering of group 2 in subgroups seems vague, there is a clear tendency toward clustering according to the calcite and moganite contents. Subgroup 2a comprises quartz (87.1–90.2 wt %) and moganite (9.3–11.1 wt %), whereas calcite is present only in minor quantities. Subgroup 2b comprises quartz (56.8–89.3 wt %) and moganite (up to 10.7 wt %), but calcite is present in higher abundances (9.5–39.6 wt %). The majority of these values represent specimens mostly collected from Kastos island, and one sample from Ithaca island. Subgroup 2c is characterized by the abundance of quartz (94.7–98.1 wt %) and minor quantities of moganite (up to 3.3 wt %) and calcite (1.7–3.1 wt %) as well (specimens from Ithaca and one from Kastos islands). The subgroup 2d, where specimens from Kefalonia island first appear, displays samples enriched in moganite (up to 25%), showing variation in the quantity of quartz and moganite contents, whereas calcite is generally absent.

## 8. Petrographical Analysis of Thin Sections

Petrographic analysis conducted on selected samples (APM: samples A12g from group 1, A7 from group 2, S1 from group 4; corresponding to the APM, and sample ITHS2 from Ithaca Island; corresponding to the IB) helped us verify the mineralogical data obtained through XRPD analysis and retrieve information about the textural features characterizing the chert nodules.

The main mineralogical constituent of the APM samples is micro- to cryptocrystalline quartz (<30 μm) with interlocking grains and irregular boundaries developing fiber chalcedony with radial extinction in forms of spherulite. They often contain anhedral-disseminated impurities, including traces of calcite and more rarely metal oxides. The petrographic analysis of the sample A12G indicates obvious differentiation patterns in the material, representing the transition from the core to the rim of the nodule (Figure 23a,b). Oval-shaped ghost areas, where micritic calcite is fully replaced by cryptocrystalline quartz, were identified in the core of the nodule, rarely maintaining relict anhedral calcite crystals surrounded by microcrystalline quartz aggregates (Figure 23c). Micritic calcite is often observed to delimit the rim from the core of the nodule, giving locally a brownish color. Fiber chalcedony spherulites were identified both in the core and the rim of the nodule, never exceeding the size of 250 μm, and were often surrounded by micritic calcite (Figure 23d). Similar textural characteristics, including compositional differentiations (Figure 24a,b) and ghost areas (Figure 24c), were also frequently observed in the sample A7, reflecting the recrystallization processes. From this thin section, the internal structure of this nodule is unclear, showing gradual compositional differentiations from the core to rim. The core in this sample seems to be slightly enriched in the fiber chalcedony spherulites (Figure 24c,d), which is verified by the XRPD analysis, characterized by bigger size (up to 500 μm). According to the semi-quantitative estimation, sample S1 contains the higher moganite content in both the

core and the rim, compared to the aforementioned samples. Through the petro-graphic study, we identify compositional layering from the core to the rim, including the presence of trace micritic calcite especially in the rim of the nodule (Figure 25a,b), abundant biogenic material forming oolites (Figure 25c), and the presence of metal oxides. In the rims, zones enriched in fiber chalcedony spherulites were observed (Figure 25d).

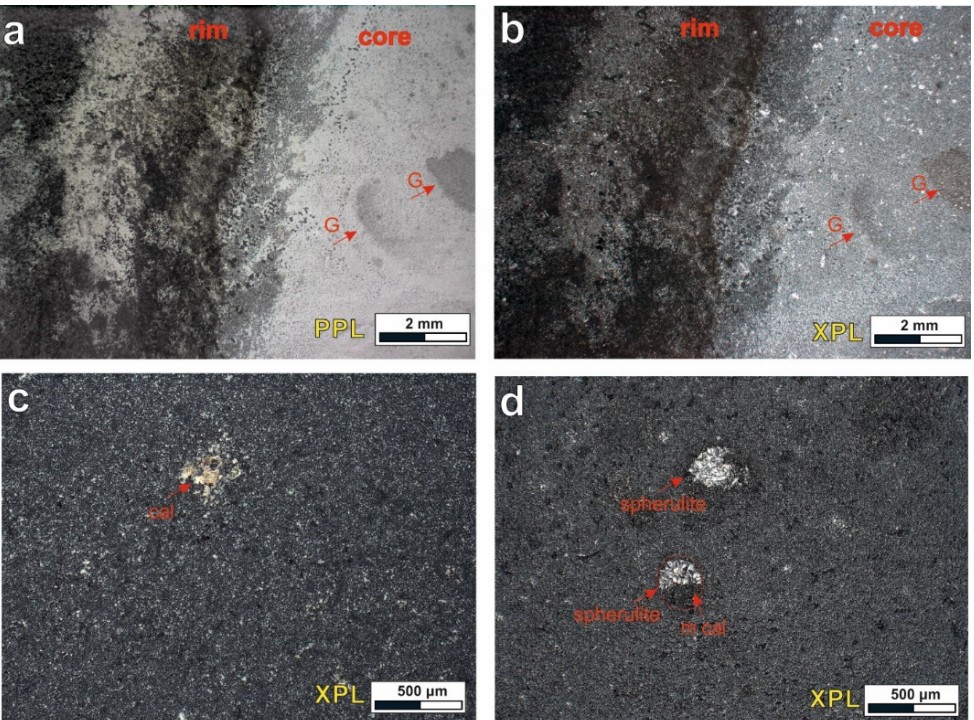

**Figure 23.** Representative photomicrographs of sample A12g. (**a**) The presence of gradual layering indicating compositional differentiation from the core to the rim of the nodule, as well as the presence of characteristic ghost-areas (G) in plane-polarized light (PPL) and (**b**) in cross-polarized light (XPL). (**c**) Microsparitic relict calcite (cal) crystal partially replaced by microcrystalline quartz, in cross-polarized light, and (**d**) the presence of fiber chal-cedony spherulites and micritic calcite (m cal), in cross-polarized light.

In the case of IB, the silica material is coarser than that observed in the case of Kastos, and even the commonly present spherulites of chalcedony fibers reach greater size, often exceeding 200 μ. The spherulitic chalcedony fibers are the first form of silicification adjacent to the chert nodules. The XRPD analysis of the sample ITHS2 indicates minor mineralogical variations between quartz and moganite, related to the area of the nodule analyzed. These mineralogical differentiations are clearly observed in the petrographic analysis as well, through compositional zoning (Figure 26a,b). Areas enriched in oolites replaced, often partially, by spherulites of radiating chalcedony fibers and ghost areas are observed (Figure 26c,d), indicating variations in silicification. Elongated silica-rich areas represent isometric mega-crystalline quartz and fibers of chalcedony that fill the primary porosity (Figure 26d).

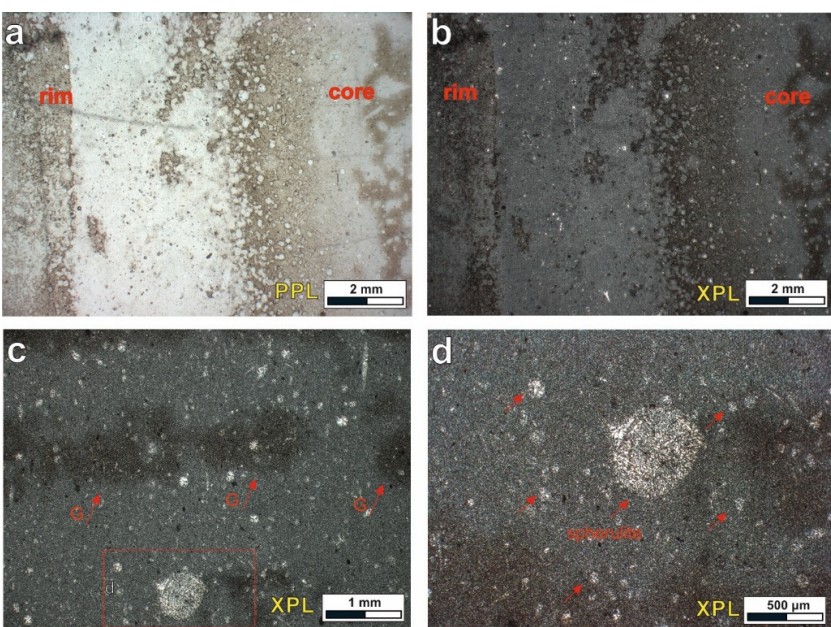

**Figure 24.** Representative photomicrographs of sample A7. (**a**) The presence of compositional differentiation with the presence of cryptocrystalline quartz and micritic calcite, disseminated irregularly in the nodule, in plane-polarized light (PPL) and (**b**) n cross-polarized light (XPL). (**c**) The presence of characteristic ghost-areas (G) in the micro- to cryprocrystalline quartzitic groundmass, in cross-polarized light, and (**d**) the presence of fiber chalcedony forming spherulites with various sizes, in cross-polarized light.

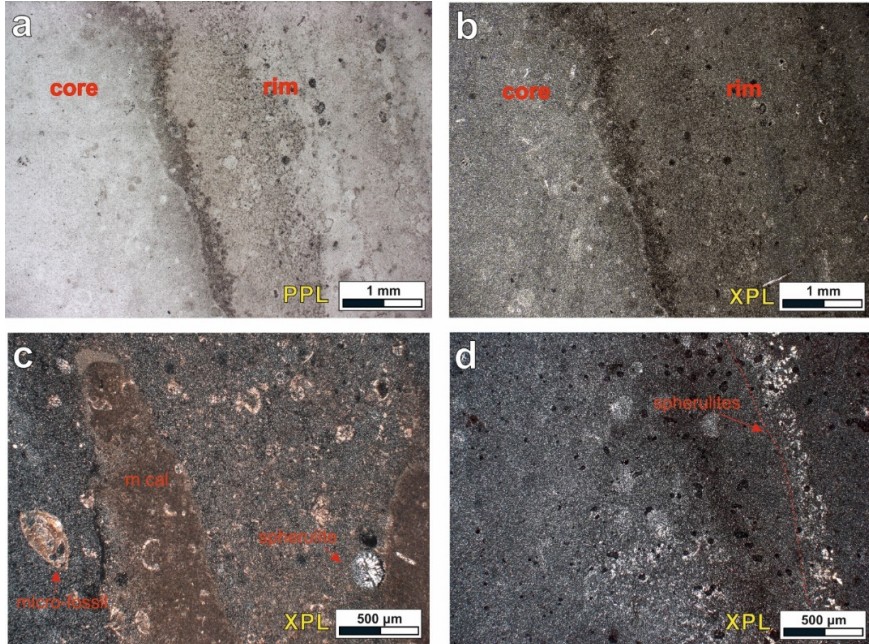

**Figure 25.** Representative photomicrographs of sample S1. (**a**) The presence of gradual layering, indicating compositional differentiation from the core to the rim of the nodule, in plane-polarized light (PPL) and (**b**) in cross-polarized light (XPL). (**c**) Areas rich in micritic calcite (m cal) and microfossils, as well as the presence of fiber chalcedony spherulites, in cross-polarized light, and (**d**) areas enriched in fober chalcedony spherulies towards the rim of the nodule, in cross-polarized light.

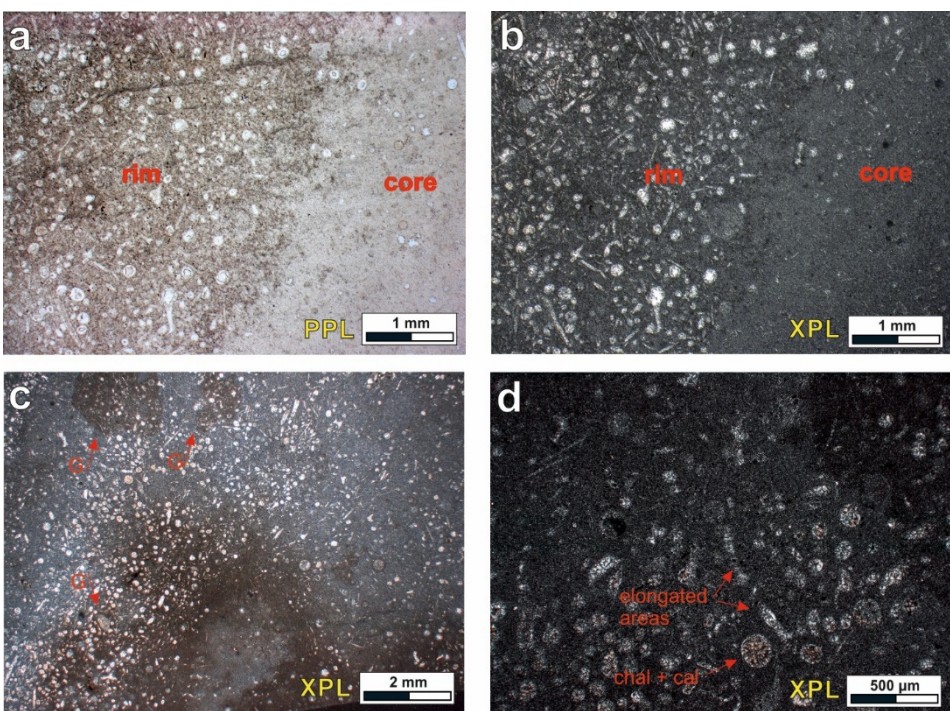

**Figure 26.** Representative photomicrographs of sample ITHS2. (**a**) The presence of gradual layering indicating compositional differentiation from the core to the rim of the nodule, in plane-polarized light (PPL) and (**b**) n cross-polarized light (XPL). (**c**) Characteristic ghost-areas (G) and spherulites in the micro- to cryptocrystalline quartz, in cross-polarized light, and (**d**) oolites replaced partially by spherulites of radiating chalcedony fibers and elongated silica-rich areas due to the primary porosity, in cross-polarized light.

## 9. Discussion

The results obtained from the mineralogical analysis of the siliceous concretions, selected from Kefalonia island, indicate that they consist mostly of quartz and moganite, while calcite either is absent or participates only in a few samples, in minor/trace quantity (Figure 27a). The only exception is specimen S3, where calcite occurs in slightly higher proportions therefore differing from all the other samples.

There are great mineralogical differences between APM and IB, siliceous concretions that could be related to many different conditions: 1. the basin shape and configuration, during rift stage of the IB, producing sub-basins with many margins (Figure 28a) [27]; 2. the underlying Triassic evaporites and the several synsedimentary faults in IB (Figure 28b) [31]; 3. the abundance of a stylolites network, which mostly characterizes the Paleocene deposits of APM, and increased the fluid flows; 4. the mineralogical composition and differences between the different geotectonic settings; and 5. the different mineralogical composition between sedimentation processes of different ages.

Siliceous concretions (nodules) are mainly characterized by the development of a core, around which a ring (rim) is formed. Mineralogical study indicates that the rim is usually richer in moganite than the core. However, nodules form Gidaki area, Ithaca seem to be an exception, being generally smaller and presenting either an internal homogeneity or non-uniform asymmetric areas of different colors with the specific mineralogical variations mentioned above, in contrast to the southern part of Ithaca. Homogeneous nodules, without a discernible inner core and outer rim, were rarely observed in Kastos and Kefalonia islands.

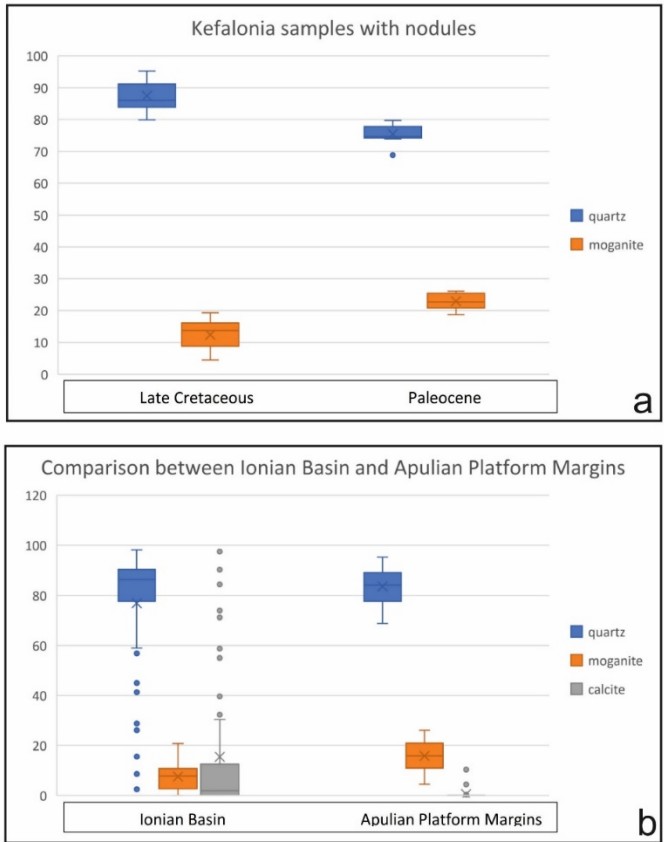

**Figure 27.** Boxplot diagrams showing the ranges defined by the semi-quantification of quartz, moganite, and calcite, differentiating (**a**) Late Cretaceous and Paleocene samples, from Kefalonia island; and (**b**) samples collected from Ithaki, Kastos, and Atokos islands, i.e., Ionian Basin compared to Kefalonia island representing the APM.

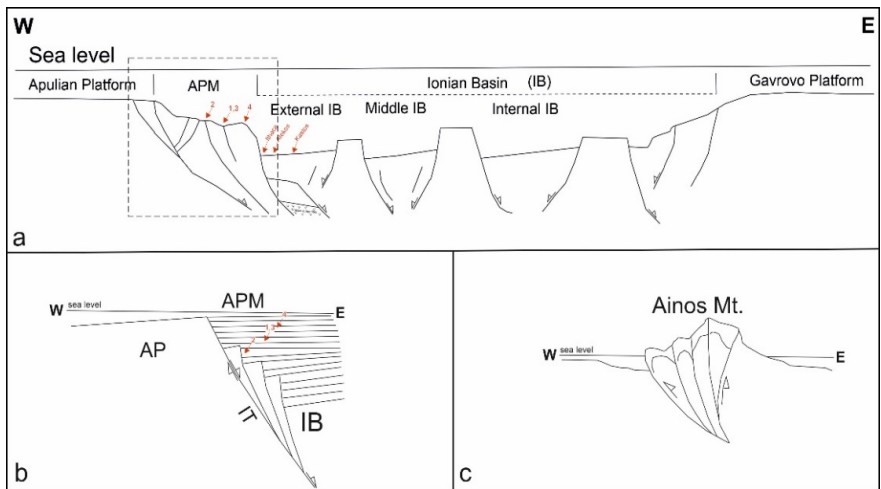

**Figure 28.** (**a**) Schematic synsedimentary cross-section, during Cretaceous to Paleocene, showing the fault-controlled APM and the sub-division of IB into troughs and highs. The studied localities are marked with red arrows, either with numbers, representing the studied groups in the APM, or the studied islands in the IB, I = Ithaca, A = Atokos, and K = Kastos. The dashed box marks the area described in detail in Figure 28b. (**b,c**) Presents the APM area in two stages; stage 1 (**b**) represents the end of the syn-rift stage sedimentation, and stage 2 (**c**) shows the final stage (current) after the tectonic regime changed from extension to compression.

The presence of halite and anhydrite, only in the Ionian basin samples, supports the role of later diagenetic fluids from the underlying deposits. Especially, in the Ionian basin, there are fault pathways for basinal fluids from the underlying Triassic evaporites that include halite and gypsum. Additionally, the presence of dolomite in the Ionian Basin specimens is in accordance with the presence of dolomitized limestones at the lower parts of the early Cretaceous "Vigla formation". Either the dolomite replaced calcite before the final growth of the siliceous bed horizons, or chert was replaced by dolomite in the manner described by [32].

One major difference between the studied samples is that the Lower Cretaceous nodules in "Vigla" limestones are abundant but they have a very small size in relation to the upper Cretaceous nodules in the calciturbidites from the Ionian Basin. The mean size of lower Cretaceous nodules is up to 6 cm, whereas that of Upper Cretaceous nodules is up to 20 cm. The different size could be related to (a) the availability of biogenic silica and (b) the stability of redox related diagenetic zones. They were stable for a longer time period in thick-bedded turbidites than in the steadily accumulating sediments. The same differences have also been recognized in Kefalonia island, where the nodules in the Cretaceous deposits are in abundance but have a very small size compared to the Paleocene nodules, which are more than one hundred (100) cm in length. The above difference in the size of nodules could be related to the huge stylolite network that has been recorded in the Paleocene deposits indicating a much more accessible primary porosity.

It should be noted that in this automated process, the 2θ values and the counts for each acquisition step, of our X-ray diffraction patterns, represent in fact a set of variables [33]. In multivariate statistical methods such as cluster analysis, especially through an automated approach, the variables should be clearly defined.

However, in our study, other parameters like instrumental settings or minor differences in the chemical composition, as well as the crystallinity of the minerals (well-crystalized or amorphous mineral phases), are likely to influence the shape of the peak positions or even shift them locally. For all these reasons, and taking into consideration the general characteristics of the whole dataset, we can conclude that cluster analyses gave satisfactory results, clustering samples according to their main mineralogical variations.

Petrographic study of thin section from the selected samples (A7, A12g, S1, and ITHS2) supports the mineralogical analysis from the XRPD. In detail, the compositional differentiations observed indicate ranging quantities quartz, moganite, and even calcite. Calcite was identified as trace through the semi-quantitative analysis only in the rims of the nodules from the samples A12g and S1. However, the pretrographic analysis also showed the presence of ghost-areas, where micritic calcite is replaced by cryptocrystalline quartz or the presence of micritic calcite mostly towards the rims. The above differences between the two methodologies were probably due to the small calcite amount, which is untraceable through the XRPD analysis.

Moreover, the great difference in moganite content between group 4 (sample S1) and the other samples, from the other groups, could also be related to the abundance in fossil fragments (Figure 25c), adding primary porosity and increasing fluid flows.

It is critical to mention that the presence of secondary fractures, both above and under the nodules, and especially towards the areas where the thickest parts of the nodules occurred, could be increased by the secondary porosity of the rocks and must be studied in detail in order to find the impact of the fractures in porosity values.

Additionally, chert nodules are characterized by microporosity, with microcrystalline quartz and the formation of joints that die out in the host limestones, producing particularly permeable rocks [15]. The above-mentioned nodules, together with faults and bedding planes in limestones, can act as fairways for fluid flows, including hydrocarbons.

Moreover, the detailed study of the stylolite network, and measurements of their shape and their development-type classification, according to previous works [13,33], is necessary to determine the genetic influences of stylolites on fluid flows, if there are any.

## 10. Conclusions

Moganite is generally abundant in all samples from Kefalonia. Specifically, Paleocene nodules seem to be more enriched in this mineral, not only compared to Cretaceous nodules from Kefalonia but also to the rest of the analyzed samples from present and previous studies.

Nodules related to IB generally show a variation between their mineralogical components in terms of quartz, moganite, and calcite content (Figure 23b), reflected often in their morphological features (i.e., core–rim).

Calcite is mostly observed in nodules collected from Kastos and Atokos islands. Particularly, the nodule from Atokos is a characteristic sample, in which the core comprises approximately equal contents of quartz and calcite and the outer core is rich in calcite, whereas quartz participates with minor quantities and its outer rim is rich in quartz, while moganite also appears.

High contents of calcite were observed in a single nodule from Ithaca, with a uniform structure and no distinguishable sub-regions. On the other hand, nodules associated with the Apulian platform margins mainly show higher contents of moganite and minor contents or no calcite.

Cluster analysis results support the previous results and can be used for further investigation. The samples are generally grouped according to the quantity of their mineralogical components, i.e., quartz, moganite, and calcite, usually successfully differentiated in the two studied geological environments.

The mineralogical differences, either between different geodynamic settings (APM and external Ionian sub-basin) or internally within the same setting, but with different ages of development, could be related to the presence-abundance of stylolites in the limestones, the later fluid flows, the restrictions from one area to the other, and the mineral composition of fluids.

The presence of secondary fractures, related to the development of nodules, must be studied in detail, as they could increase the secondary porosity values.

**Supplementary Materials:** The following are available online at https://www.mdpi.com/article/10.3390/min11080890/s1, Table S1: Coordinate samples.

**Author Contributions:** Conceptualization, N.B.; methodology, M.K., I.I. and G.I.; software, N.D.; formal analysis, N.B. and G.I.; investigation, N.B., M.K., G.I., I.I. and A.Z.; data curation, E.Z.; writing—original draft preparation, N.B., M.K., G.I. and A.Z.; writing—review and editing, N.B. and A.Z.; supervision, A.Z. All authors have read and agreed to the published version of the manuscript.

**Funding:** This research received no external funding.

**Acknowledgments:** We would like to thank the reviewers for their comments, which improved the final version.

**Conflicts of Interest:** The authors declare no conflict of interest.

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
