# Peer review of "Comparison between Siliceous Concretions from the Ionian Basin and the Apulian Platform Margins (Pre-Apulian Zone), Western Greece: Implication of Differential Diagenesis on Nodules Evolution"

_minerals, doi:10.3390/min11080890_

Round 1
Reviewer 1 Report
The manuscript entitled "Comparison between siliceous concretions from the Ionian Basin and the Apulian Platform margins (Pre-Apulian zone), western Greece: implications for the different diagenetic processes on evolutionary conditions" submitted by Bourli and co-authors provides a nice dataset of outcrop observations and mineralogical analysis of siliceous concretions in the pre-apulian zone and ionian basin.
The manuscript however shows several critical weaknesses that force me to ask for major revisions.
The main issue is the presentation of the results, which is really confusing. I suggest the authors to rexork seriously the structure of the paper, by presenting first the outcrop observations, then the samples, then the lab results (XRD).
The discussion is missing some maturity and illustrative content, such as a paragenetic sequence established from the observation, a basin / local conceptual scheme of flow pathways. Conclusions have to be reworked when the results and discussion parts will be revised.
The topic is of broad interest, and thus, I would really recommend the authors to improve this manuscript for publication.
The authors may find attached the annotated manuscript for further detailed corrections.

Author Response
The revised paper was re-organized, as suggested, with clear presentation first of the outcrops (chapter 4), then describing in a new chapter the selected concretions from outcrops and after cutting them in the lab. (chapter 5). After the above two chapters with the field work, there is a chapter with mineralogical analysis (chapter 6), cluster analysis in chapter 7 for the comparisons between previous work and present studies, and a new chapter with the petrographical analysis of thin sections (new chapter 8).
In detail and according the PDF file and the marks on it.
- The title changed as suggested
- Lines 24-29 from abstract re-organized as suggested.
- References were added in the introduction chapter as suggested.
- In introduction chapter lines 49-50, 55-56, 62-65, 77-78 rephrased.
- Geological setting chapter was re-organized as suggested, some paragraphs moved in other places or some descriptions improved.
- Chapter methodology re-organized and some details were added, as suggested.
- In chapter 4 and for lines 190-191, the sample positions added on the maps.
- The Table 1 was cutted into two tables, as suggested.
- In relation with the comment on lines 326-328: “please explain why no quantitative estimation of the mineralogical content is provided.” This work is a comparison work with previous published one from Bourli et al., 2019. Therefore, we followed the same way in order to be comparable for the readers.
- Chapters’ Discussion and conclusions re-organized, new material was added and cluster analysis moved in an independent chapter.
Reviewer 2 Report
This manuscript concerns a problem, that could be really interesting, if properly investigated and presented well. The siliceous concretions in carbonate-bearing rocks may be an exciting object of studies and a source of informations on the sediment accumulation and on diagenetic and epigenetic processes.
However I am sorry to write, that this manuscript has not features of a good scientific report. First, the description of the field investigations is chaotic. Next, the macroscopic characteristic of the concretions is perfunctory and the photographs are mostly of poor quality and do not enable the reader an undoubted observation of the presented concretions. It happens that the explanation of the image suggests something else than it may be seen in the picture (explanation – absence of zoning, picture – zoning visible, etc.). The laboratory investigation included (only) the determination of contents of mineral components of the concretions by the XRPD method with grouping the results by a computer program. This led to the discussion and conclusions, which were in fact considerations “either – or” and suppositions. But even simple optic microscope observations (not made by the authors) in such case give valuable data being the source of the reasonable genetic reconstruction. I even do not dare to suggest other methods.
The data obtained by the authors, if elaborated properly, would be sufficient to prepare few-page communication, not an almost 25 pages review.
Moreover, the text has numerous mistakes – linguistic, stylistic and of substantive incorrectnesses. Especially the use of the first letters of the words as capital ones is chaotic and frequently senseless. Some used words like “geometry” or “methodology” introduce a sound of jargon, whereas the scientific text should be precise. My individual remarks are added to the manuscript pdf file as sticky notes.
The manuscript in its present form is not suitable to publication. Also the text should be carefully improved by an English native speaker of geological knowledge.

Author Response
Eight are the general points that reviewer 2 was mentioned and mostly related with ms organization:
This manuscript concerns a problem, that could be really interesting, if properly investigated and presented well. The siliceous concretions in carbonate-bearing rocks may be an exciting object of studies and a source of informations on the sediment accumulation and on diagenetic and epigenetic processes. However, I am sorry to write, that this manuscript has not features of a good scientific report.
- First, the description of the field investigations is chaotic. MS re-organized as suggested and from other reviewer
- Next, the macroscopic characteristic of the concretions is perfunctory and the photographs are mostly of poor quality and do not enable the reader an undoubted observation of the presented concretions. It happens that the explanation of the image suggests something else than it may be seen in the picture (explanation – absence of zoning, picture – zoning visible, etc.). Photographs and their caption was improved, but we disagree with the comment about the quality of the photographs because their quality was excellent. If you would like we can ask from the journal to have access in the original high quality figures.
- The laboratory investigation included (only) the determination of contents of mineral components of the concretions by the XRPD method with grouping the results by a computer program. Yes, when you want to run a comparison between different areas or results of this study with previous published results you have to find a method and for us was “cluster analysis”. After the above, reviewer 2, point we re-organized the ms including a separate chapter, this of cluster analysis.
- This led to the discussion and conclusions, which were in fact considerations “either – or” and suppositions. Both chapters re-organized
- But even simple optic microscope observations (not made by the authors) in such case give valuable data being the source of the reasonable genetic reconstruction. I even do not dare to suggest other methods. We added a new chapter with petrographical analysis from thin sections, as suggested.
- The data obtained by the authors, if elaborated properly, would be sufficient to prepare few-page communication, not an almost 25 pages review.
- Moreover, the text has numerous mistakes – linguistic, stylistic and of substantive incorrectnesses. Especially the use of the first letters of the words as capital ones is chaotic and frequently senseless. Some used words like “geometry” or “methodology” introduce a sound of jargon, whereas the scientific text should be precise. My individual remarks are added to the manuscript pdf file as sticky notes. We accepted all changes, moreover the English language was checked, and many changes were introduced from a native researcher. Additionally, and although we disagree with the general comments of the reviewer concerning the used terms like “geometry” and “methodology” that introduce a sound of jargon we tried to avoid any collision with the reviewer and we followed his/her suggestions.
- The manuscript in its present form is not suitable to publication. Also the text should be carefully improved by an English native speaker of geological knowledge.
In detail and according the PDF file and the marks on it.
- We accepted most of the reviewer suggestions and comments on his/her PDF file, in order to improve our ms.
- The title changed as suggested.
- Lines 34, 221, 227, 250, e.t.c. where for reviewer is not accepted the term “bedding” or “bedding plane”, we disagree, because this is an international term showing or describing the bedding surface or the fault plane.
- In lines 34-35 and in relation with the comment about the deep-water margins. Yes, we have deep-water margins.
- We accepted the points with the term “morphology” and we changed it as suggested to “shape”.
- We disagree with the comment on [“Geometry" is a huge part of science; here shape (and arrangement?), "geometries" sound like a jargon], both for the basin and for the stylolites. Yes, we are talking for the geometry of the basin (symmetrical or asymmetrical, elongated or restricted, e.t.c., or the geometry of the stylolites, concerning their development and their distribution. We agree that it is better to use the term “shape” and not “geometry” when we describe a gravel or a clast, and for this reason we accepted this point and we changed the used term.
- We changed all letters of “lower” or “upper” to “Lower” and “Upper” both in the text and in the tables.
- We changed all used capital letters within the text like “Moganite”.
- Figure 2 captions improved, according reviewer suggestions.
- In figure 21 and in relation to the question mark on line 343, probably it was not so clear in order to understand that this figure include seven diffractograms and not one. On the right end of each diffractogram there is the number of each analyzed sample (ATOK_1, ATOK_2, ATOK_3 e.t.c.
Reviewer 3 Report
The work of Bourli et al. presents an interesting comparison between siliceous concretions from the Ionian Basin and the Apulian Platform margins in western Greece. However, before the publication, it needs to be addressed the following issues:
Some figures (8a, 9, 11, 12) lack the scale and should be added.
The PDF (ICDD) file numbers of the identified phases by PXRD should be included in the text.
There are XRD peaks that are not indexed (fig. 21)? Are there any other phases?
Using the Rietveld method, the authors should check the estimated (by cluster analysis) amounts of quartz, calcite, and moganite.
Page 20. Please avoid using "...very low quartz content.", "Moganite is present with significant contents...." without mentioning any numbers.
Fig. 23. Please indicate the meaning of the Y-axis.
Author Response
The following points highlighted by reviewer 3:
The work of Bourli et al. presents an interesting comparison between siliceous concretions from the Ionian Basin and the Apulian Platform margins in western Greece. However, before the publication, it needs to be addressed the following issues:
- Some figures (8a, 9, 11, 12) lack the scale and should be added. Improved – done.
- The PDF (ICDD) file numbers of the identified phases by PXRD should be included in the text. Included – dome.
- There are XRD peaks that are not indexed (fig. 21)? Are there any other phases? No, there are no picks that have not been recognized. We did not label only the small reflections, however now we have labelled them all.
- Using the Rietveld method, the authors should check the estimated (by cluster analysis) amounts of quartz, calcite, and moganite. We tried to compare our results with results from Bourli et al. 2019. Therefore, and in order to be directly comparable, we followed the same process and methodology.
- Page 20. Please avoid using "...very low quartz content.", "Moganite is present with significant contents...." without mentioning any numbers. The contents have been explained in detail in the revised text, as well as in the corresponding table.
- 23. Please indicate the meaning of the Y-axis. Done
Round 2
Reviewer 2 Report
The present manuscript is much better than its earlier version.
The title is still a bit “baroque” though shorter. I’d rather suggest first to indicate “what”, and afterwards “where”, e.g.:
Implication of different diagenesis on siliceous nodules evolution in the Ionian Basin and the Apulian Platform margins (Pre-Apulian zone), western Greece;
“comparison” is obsolete, because it is evident, that studies of two or more objects include comparison of the results.
The style of the manuscript is much improved and during reading there is now an impression of real scientific text. The illustrations, especially photos, are of much better quality and the figure explanations are well prepared.
I have no other remarks.